# A PRISMA systematic review of adolescent gender dysphoria literature: 1) Epidemiology

**Lucy Thompson**[1,2,3]*, **Darko Sarovic**[1], **Philip Wilson**[3], **Angela Sämfjord**[4], **Christopher Gillberg**[1,2]

**1** Gillberg Neuropsychiatry Centre (GNC), Department of Psychiatry and Neurochemistry, Institute of Neuroscience and Physiology, Sahlgrenska Academy, University of Gothenburg, Göteborg, Sweden, **2** Institute of Health and Wellbeing, University of Glasgow, Glasgow, United Kingdom, **3** Institute of Applied Health Science, Centre for Health Science, University of Aberdeen, Inverness, United Kingdom, **4** The Child and Adolescent Psychiatric Clinic, The Queen Silvia Children's Hospital, Gothenburg, Sweden

* Lucy.Thompson@gnc.gu.se

**Data Availability Statement:** All data are provided within the article.

**Funding:** The author(s) received no specific funding for this work.

## Abstract

It is unclear whether the research literature on adolescent gender dysphoria (GD) provides sufficient evidence to adequately inform clinical decision making. In the first of a series of three papers, this study sought to systematically review published evidence regarding: the prevalence of GD in adolescence; the proportions of natal males/females with GD in adolescence and whether this changed over time; and the pattern of age at (a) onset (b) referral and (c) assessment. Having searched PROSPERO and the Cochrane library for existing systematic reviews (and finding none), we searched Ovid Medline 1946 –October week 4 2020, Embase 1947–present (updated daily), CINAHL 1983–2020, and PsycInfo 1914– 2020. The final search was carried out on the 2nd November 2020 using a core strategy including search terms for 'adolescence' and 'gender dysphoria' which was adapted according to the structure of each database. Papers were excluded if they did not clearly report on clinically-verified gender dysphoria, if they were focused on adult populations, if they did not include original data (epidemiological, clinical, or survey) on adolescents (aged at least 12 and under 18 years), or if they were not peer-reviewed journal publications. From 6202 potentially relevant articles (post de-duplication), 38 papers from 11 countries representing between 3000 and 4000 participants were included in our final sample. Most studies were observational cohort studies, usually using retrospective record review (26). A few compared to normative or population datasets; most (31) were published in the past 5 years. There was significant overlap of study samples (accounted for in our quantitative synthesis). No population studies are available, so prevalence is not possible to ascertain. There is evidence of an increase in frequency of presentation to services, and of a shift in the natal sex of referred cases: those assigned female at birth are now in the majority. No data were available on age of onset. Within the included samples the average age was 13 years at referral, 15 years at assessment. All papers were rated by two reviewers using the Crowe Critical Appraisal Tool v1·4 (CCAT). The CCAT quality ratings ranged from 45% to 96%, with a mean of 78%. Almost half the included studies emerged from two treatment centres: there was considerable sample overlap and it is unclear how representative these are of the adolescent GD community more broadly. The increase in clinical presentations of GD,

**Competing interests:** The authors have declared that no competing interests exist.

particularly among natal female adolescents, warrants further investigation. Whole population studies using administrative datasets reporting on GD / gender non-conformity may be necessary, along with inter-disciplinary research evaluating the lived experience of adolescents with GD.

## Introduction

Gender Dysphoria (GD) is a categorical diagnosis in the Fifth Edition of the Diagnostic and Statistical Manual of Mental Disorders (DSM-5) [1]. It is also used as a general descriptive term referring to a person's discontent with assigned gender. In recent years, GD diagnoses have been increasingly made in child and adolescent services [2–4]. There has been a parallel increase in demand for gender transition interventions, particularly among natal females [2–4], and including pre-adolescents [5]. Such transitions have increasingly involved the use of puberty suppression, cross-sex hormones and surgical procedures, usually in accordance with the so-called 'Dutch model', where intervention is staged in accordance with a young person's age and stage of pubertal development [6]. Calls to improve availability of medical interventions have sometimes been made on the basis of reports of much increased levels of mental health problems, including suicide attempts, among youth with GD [7, 8], and claims that the medical procedures referred to would improve mental health outcomes [6, 9].

Gender and sex are two terms which are often used interchangeably but that are not synonymous. The sex of a person refers to either male or female status broadly based on the sex chromosomes and genitalia. The word is used in social, medical and legal contexts in most countries to 'categorise' people under the two sexes, as boys/men or girls/women.

The term gender is harder to define, as it reflects how the individual identifies or feels, how a person 'fits' with social norms, activities and attributes that are commonly associated with male or female sex. For many people sex and gender are consistent, but for some, there is discordance between the biological anatomy of the body and gender self-perception/identity. The term 'transgender' is often used to describe this identity, whilst others will relate to terms such as GD or gender incongruence. The literature and common understanding of this area is evolving extremely rapidly, and there is increasing acknowledgement that gender identity may not relate to a binary gender definition at all. Gender does not necessarily reflect sexual orientation, an enduring pattern of emotional, romantic and/or sexual attraction [10]. Gender identity on the other hand is a component of one's personal multi-dimensional sense of self, encompassing moral, ethical, spiritual / religious beliefs [11]. For most people who do not have abnormalities of the external genitalia, sex is documented at birth and, in the majority population, sex and gender are consistent throughout life.

A strict definition, such as in the DSM-5 [1], requires that individuals diagnosed with GD have to suffer clinically significant distress or impairment in social, school, or other important areas of functioning. A range of terminology is used to describe young people experiencing GD. It is apparent from the literature that the terminology is changing rapidly, and that the terms 'assigned female / male at birth' (AFAB /AMAB) or 'natal fe/male' are commonly used in recent literature. We have chosen to use the term 'natal fe/male' (abbreviated to NF / NM) because it is less cumbersome and is inclusive of those experiencing GD and who not identify with either male or female genders (although we acknowledge it excludes many intersex people).

There is currently an intense international debate regarding a number of issues relating to GD [12]. A recent high profile example involved the Gender Identity Development Service

(GIDS) in London, UK, altering its procedures due to a High Court ruling that it would be 'highly unlikely' that children under 13, and 'very doubtful' that 14- and 15-year olds, could be Gillick competent [13], and therefore they could not consent to puberty suppression treatment [14]. This decision was met with equivocal support and criticism, not least as it has implications for the broader application of the Gillick framework for consent to medical procedures. (It has since been successfully appealed [15]) A lack of good quality evidence has been acknowledged [16]. A recent review by the Swedish Agency for Health Technology Assessment and Assessment of Social Services [17] indicated that there is very little in terms of empirical evidence in the field, both in terms of overall GD epidemiology, the association of GD with mental health problems, the rate and types of medical interventions provided and outcomes (including outcomes for those not treated medically or surgically) in the longer term.

## Scope of the review

This review addresses the first of three sets of questions addressing the current state of evidence on gender dysphoria experienced in adolescence. Our over-arching aim was to establish 'what does the literature tell us about gender dysphoria in adolescence?' We broke this down into seven specific questions:

1. What is the prevalence of GD in adolescence?

2. What are the proportions of natal males / females with GD in adolescence (a) and has this changed over time (b)?

3. What is the pattern of age at (a) onset (b) referral (c) assessment (d) treatment?

4. What is the pattern of mental health problems in this population?

5. What treatments have been used to address GD in adolescence?

6. What outcomes are associated with treatment/s for GD in adolescence?

7. What are the long-term outcomes for all (treated or otherwise) in this population?

The present paper focuses on questions 1, 2, 3a, 3b, and 3c. We shall address question 4 in a second paper, and questions 3d, and 5–7 in a final paper. The methodology below includes the searches conducted for the whole review.

We set out to include any paper offering primary data in response to any of these questions, regardless of the focus of that paper.

## Methods

### Protocol and registration

The systematic review protocol was submitted to PROSPERO on the 28[th] November 2019, and registered on 17 March 2020 (registration number CRD42020162047). An update was uploaded on 2[nd] February 2021 to include specific detail on age criteria and clinical verification of condition. The review has been prepared according to PRISMA 2020 [18] guidelines (see S1 Checklist).

### Eligibility criteria

The volume of non-peer-reviewed literature in initial searches proved so great that we took the decision to only include peer-reviewed journal papers featuring original research data. This decision was made subsequent to initial PROSPERO registration, but prior to full text screening. Complete inclusion criteria were:

- Focused on gender dysphoria or transgenderism;

- Includes data on adolescents (aged 12–17 years inclusive);

- Includes original data (not review paper or opinion piece);

- Peer-reviewed publication (not theses or conference proceedings);

- In English language.

### Information sources

We searched PROSPERO and the Cochrane library for existing systematic reviews. We searched Ovid Medline 1946 –October week 4 2020, Embase 1947–present (updated daily), CINAHL 1983–2020, and PsycInfo 1914–2020. After selecting the final sample of articles, the first author used their reference lists as a secondary data source.

### Search

The final search was carried out on the 2nd November 2020 using a core strategy which was adapted according to the structure of each database. The core strategy included search terms for 'adolescence' and 'gender dysphoria'. The specific search strategies employed in each database are detailed in Table 1.

**Table 1. Search terms.**

| | EMBASE | Ovid Medline | CINAHL | PsycInfo |
|---|---|---|---|---|
| **Adolescence** | 1 Exp adolescence/<br>2 (adolesc* or teen* or puberty*).tw. | 1 Exp adolescence/<br>2 (adolesc* or teen* or puberty*).tw. | 1 (MH "Adolescence+")<br>2 TI adolesc* OR TI teen* OR TI pubert* OR AB adolesc* OR AB teen* OR AB pubert* | 1 TI adolescence OR AB dolescence<br>2 TI adolesc* OR TI teen* OR TI pubert*<br>3 AB adolesc* OR AB teen* OR AB pubert* |
| **Gender Dysphoria** | 3 exp gender dysphoria/<br>4 exp transgender/<br>5 sex reassignment/<br>6 (gender dysphoria or gender identity or transsex* or trans sex or transgender or trans gender or sex reassignment).tw. | 3 exp gender dysphoria/<br>4 exp transgender/<br>5 Exp Sex Reassignment Procedures/<br>6 (gender dysphoria or gender identity or transsex* or trans sex or transgender or trans gender or sex reassignment).tw. | 3 (MH "Gender Dysphoria")<br>4 (MH "Transgender Persons") OR (MH "Transsexuals")<br>5 (MH "Sex Reassignment Procedures+")<br>6 TI gender dysphoria OR AB gender dysphoria OR TI gender identity disorder OR AB gender identity disorder OR TI transsex* OR AB transsex* OR TI trans sex* OR AB trans sex* OR TI transgender OR AB transgender OR TI trans gender OR AB trans gender<br>7 TI sex reassignment OR AB sex reassignment OR TI gender reassignment OR AB gender reassignment | 4 DE "Gender Dysphoria" OR DE "Gender Nonconforming" OR DE "Gender Reassignment" OR DE "Gender Identity" OR DE "Transsexualism" OR DE "Transgender"<br>5 TI gender dysphoria OR TI gender identity disorder OR TI transsex* OR TI trans sex* OR TI transgender OR TI trans gender OR TI sex reassignment OR TI gender reassignment<br>6 AB gender dysphoria OR AB gender identity disorder OR AB transsex* OR AB trans sex* OR AB transgender OR AB trans gender OR AB sex reassignment OR AB gender reassignment |
| **Combination terms** | 7 1 OR 2<br>8 3 OR 4 OR 5 OR 6<br>9 7 AND 8 | 7 1 OR 2<br>8 3 OR 4 OR 5 OR 6<br>9 7 AND 8 | 8 1 OR 2<br>9 3 OR 4 OR 5 OR 6 OR 7<br>10 8 AND 9 | 7 1 OR 2 OR 3<br>8 4 OR 5 OR 6<br>9 7 AND 8 |

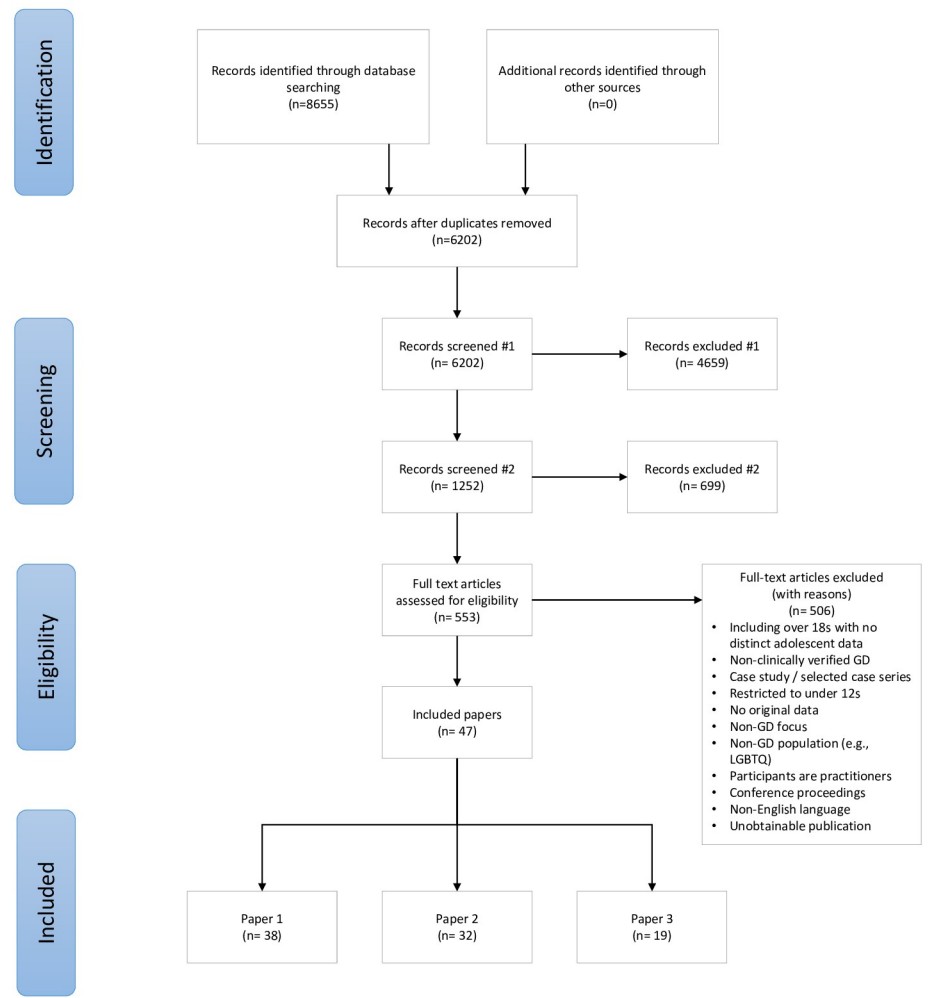

**Fig 1. PRISMA diagram.**

## Study selection

The study selection process is illustrated in Fig 1.

In the first stage of screening, papers were excluded based on their title or abstract if they did not clearly report on gender dysphoria or transgenderism and if they were focused on adult populations. In the second stage of screening, papers were excluded on the basis of title and abstract if they did not include original data (epidemiological, clinical, or survey) on adolescents (aged at least 12 and under 18 years). At both stages papers were retained if there was insufficient information to exclude them.

Full-text files were obtained for the remaining records.

Papers were rejected at this stage if they:

- Contained no original data (including literature and clinical reviews, journalistic / editorial pieces, letters and commentaries);

- Included only case studies or selected case series;

- Pertained to conditions other than GD (e.g., Disorders of sexual development or HIV);

- Did not include clinically-identified GD (e.g., survey where participants self-identify, with no clinical contact);

- Pertained to populations other than those with GD (e.g., LGBTQ more broadly);

- Pertained to populations including or restricted to those aged 18 years or older. This included papers where adolescents and adults were included in the same sample, but adolescents were not separately reported (in many cases age range was not reported and so a 'balance of probabilities' assessment had to be made based on the reported mean age);

- Pertained to populations restricted to those aged under 12 years of age. This included papers where adolescents and children were included in the same sample, but the majority of participants were clearly under 12 (based on mean or median age);

- Where participants were practitioners, not patients;

- Referred only to conference proceedings;

- Were written in a non-European language (e.g., Turkish);

- Could not be obtained (including due to being published in non-English language journals, or in theses).

Following initial full text screening, all remaining papers were assessed by a second reviewer to reduce the risk of inclusion bias. Where reviewers reached a different conclusion, discussion took place to reach consensus. If agreement could not be reached, a third reviewer was consulted, and discussion used to reach consensus amongst all three reviewers.

Data extracted from eligible papers were tabulated and used in the quantitative and qualitative synthesis. The following information was recorded: sample size; natal sex; age (years); dates of data collection; study design; study location. Given the limited number of specialist treatment centres globally, we assessed how many of the included papers featured the same or overlapping samples.

## Quality assessment

All papers were rated by two reviewers using the Crowe Critical Appraisal Tool v1·4 (CCAT [19]). CCAT is suitable for a range of methodological approaches, assessing papers in terms of eight categories: Preliminaries (overall clarity and quality); Introduction; Design; Sampling; Data collection; Ethical matters; Results; Discussion. Each category is rated out of 5 and all eight categories summed to give a total out of 40 (converted to a percentage). In the present review, each paper was then assigned to one of five categories, based on the average rating of the reviewers, where a rating of 0–20% was coded 1 (poorest quality), and 81–100% coded 5 (highest quality). Inter-rater reliability was shown to be very good ($k = 0·93$, SE = 0·05).

## Data collection process

Data were extracted from the papers using the CCAT form (https://conchra.com.au/wp-content/uploads/2015/12/CCAT-form-v1.4.pdf) by two reviewers per paper and compiled by the first author (LT). Any data missing from forms was extracted by LT. Once compiled, instances of overlap between papers (i.e., if the same sample was described in two papers) were identified and tabulated, and the final sample for each question defined.

## Results

### Number of studies included, retained and excluded

The PRISMA diagram in Fig 1 provides details of the screening and exclusion process. The searches returned 8655 results, reduced to 6202 following de-duplication. Titles and abstracts were screened by one reviewer (LT) and 4659 records excluded after initial screening and a further 699 excluded on second stage title / abstract screening. This left 553 eligible for full text screening. An initial screening (LT) of full texts reduced the number of records to 155. Forty-eight papers were included in the final dataset, of which 38 included data for the present paper. Full characteristics of included studies are provided in Table 2.

The majority of samples were from the Netherlands (n = 10), followed by the USA (n = 10), the UK (n = 7), Canada (n = 5), Belgium (n = 2), Finland (n = 2), Germany (n = 1), Israel (n = 1), Australia (n = 1), Italy (n = 1), Switzerland (n = 1) and Turkey (n = 1) (note two papers together described six samples, hence the total is 42). The Netherlands data all pertained to the same centre and research group. All seven of the UK samples came from the same Gender Identity Development Service (GIDS: Tavistock & Portman NHS Trust) in London, and three of the Canadian papers came from the same Transgender Youth Clinic in Toronto. Accordingly, not all 42 samples are necessarily mutually exclusive. Overlapping samples were not always acknowledged, and so where overlap may have occurred (based on location, setting, age and date variables) this has been noted and has been taken into account in any analysis. Fig 2 provides a graphical representation of overlap between samples and indicates which papers contributed data to which analyses. Based on the reported information, in total we estimate between 3000 and 4000 adolescents assessed at specialist centres for GD between 1980 and 2019 were included in the 38 papers.

Most studies were observational cohort studies, usually using retrospective record review (n = 26). A few studies included comparison to a normative sample or given population norms (n = 6). All but one paper was published within the past ten years (2011 or later) and all but seven in the past five years (2016 or later). Only five papers explicitly included data from before 2000 (a further six may have included pre-2000 data but did not report dates). All papers included both NM and NF participants, all studies reported the proportion of NM and NF participants in their sample, and most included age data (with age at assessment being the most widely reported) (see Table 2 and Fig 2).

Twenty-four samples were reported to have met clinical diagnostic criteria for GD / GID, usually using one of the DSM manuals (5/28 did not state which criteria were applied). The remaining 18 samples did not report whether participants met diagnostic criteria, but were included on the basis of being established patients within a specialist treatment centre, either in active assessment or treatment (n = 14) or were the result of secondary data mining where ICD 9/10 codes and appropriate keywords were used to establish likely GD (n = 4).

A substantial group of papers narrowly missed inclusion criteria, mostly on the age criterion and some on the verified GD criterion, and were not included in the final sample of reviewed papers. We documented characteristics of all studies excluded at the final full text screen in Table 5.

### Overall findings based on included studies

**1. What is the prevalence of GD in adolescence?.**   It is not possible to address this question from the existing literature. Whilst a number of surveys exist that would allow one to make estimates of prevalence, none were conducted using whole population samples. Further, we chose to focus this review only on papers where GD had been clinically verified as we were

**Table 2. Study characteristics.**

| ID | Country | Reference | Design | Setting | N | Age (years) | Male natal sex (%) | Date range | GD status |
|---|---|---|---|---|---|---|---|---|---|
| 1 | Australia | Mahfouda, et al. (2019). Mental Health Correlates of Autism Spectrum Disorder in Gender Diverse Young People: Evidence from a Specialised Child and Adolescent Gender Clinic in Australia. Journal of Clinical Medicine, 8(10), 20 [27] | obs, retro, x-sect | Gender Diversity Service (GDS), Perth Children's Hospital. GENTLE cohort study | 104 | 14·6±1·7 | 24·0 | Nov 2017-Jun 2019 | 4 |
| 2 | Belgium | Tack, et al. (2018). Proandrogenic and Antiandrogenic Progestins in Transgender Youth: Differential Effects on Body Composition and Bone Metabolism. Journal of Clinical Endocrinology and Metabolism, 103(6), 2147–2156 [34] | obs, retro, longit, interv | Division of Pediatric Endocrinology, Ghent University* | 65 | Mean NF: 16·2±1·05 Mean NM:16·3 ±1·21 | 32·3 | 2011–2017 | 1 |
| 3 | Canada | Chiniara, et al. (2018). Characteristics of adolescents referred to a gender clinic: Are youth seen now different from those in initial reports? Hormone Research in Paediatrics, 89(6), 434–441 [22] | obs, retro, x-sect | Transgender Youth Clinic (TYC), The Hospital for Sick Children, Toronto | 203 | 12–18 (mean 16) | 23·2 | 2014–2016 | 1 |
| 4 | Canada | Feder, et al. (2017). Exploring the association between eating disorders and gender dysphoria in youth. Eating Disorders, 25(4), 310–317 [35] | obs, retro, x-sect | Gender Diversity Clinic at a Canadian tertiary pediatric care hospital in Ottawa, Ontario | 97 | 12–18 | 38·1 | Oct 2007-Jul 2015 | 1 |
| 5 | Canada | Heard, et al. (2018). Gender dysphoria assessment and action for youth: Review of health care services and experiences of trans youth in Manitoba. Paediatrics & Child Health (1205–7088), 23(3), 179–184 [25] | obs, retro, x-sect | Manitoba Gender Dysphoria Assessment and Action for Youth (GDAAY) program | 174 | 4·7–17·8 | 29·9 | None given | 2 |
| 6 | Canada | Sorbara, et al. (2020). Mental Health and Timing of Gender-Affirming Care. Pediatrics, 146 (4), 10 [36] | obs & comp, retro, x-sect | Transgender Youth Clinic (TYC), The Hospital for Sick Children, Toronto | 300 | 10·5–17·9 | 23·7 | initial visit Oct 2013—Jun 2016 (cohort 1) or Aug 2017—Jun 2018 (cohort 2) | 1 |
| 7 | Finland | Kaltiala-Heino, et al. (2015). Two years of gender identity service for minors: Overrepresentation of natal girls with severe problems in adolescent development. Child and Adolescent Psychiatry and Mental Health, 9 (1) [37] | obs, retro, x-sect | Tampere University Hospital, Department of Adolescent Psychiatry | 47 | mean NM 16·04±0·57, NF 16·66 ±1·07 | 12·8 | 2011–2013 | 2 |
| 8 | Finland | Kaltiala-Heino, et al. (2019). Sexual experiences of clinically referred adolescents with features of gender dysphoria. Clinical Child Psychology and Psychiatry, 24(2), 365–378 [38] | obs, retro, x-sect | Tampere University Hospital, Department of Adolescent Psychiatry | 99 | 14–18 | 15·2 | 2011–2015 | 2 |

(*Continued*)

**Table 2.** (Continued)

| ID | Country | Reference | Design | Setting | N | Age (years) | Male natal sex (%) | Date range | GD status |
|---|---|---|---|---|---|---|---|---|---|
| 9 | Germany | Levitan, et al. (2019). Risk factors for psychological functioning in German adolescents with gender dysphoria: poor peer relations and general family functioning. European Child and Adolescent Psychiatry [39] | obs, prosp, x-sect | Hamburg Gender Identity Service for Children and Adolescents | 180 | Mean 15·5 ±1·4 | 18·9 | Sept 2013-Jun 2017 | 1 |
| 10 | Israel | Amir, et al (2020). Fertility preservation rates among transgender women compared with transgender men receiving comprehensive fertility counselling. Reproductive Biomedicine Online, 41(3), 546–554 [40] | obs, retro, x-sect | Gender Dysphoria Clinic at Dana-Dwek Children's Hospital, Gender Clinic, Tel Aviv Sourasky Medical Center | 30 | <18 years | 20 | Jan 2017-Apr 2019 | 1 |
| 11 | Italy | Fisher, et al. (2017). Psychological characteristics of Italian gender dysphoric adolescents: a case-control study. Journal of Endocrinological Investigation, 40(9), 953–965 [41] | obs & comp, prosp, x-sect | The Sexual Medicine and Andrology Unit of the University of Florence and the Gender Clinics of Rome, Milan, and Naples University Hospitals | 46 (+ 46 control) | <18 | 43·5 | Sept 2014-Feb 2016 | 1 |
| 12 | Multi | de Graaf, et al. (2018). Psychological functioning in adolescents referred to specialist gender identity clinics across Europe: A clinical comparison study between four clinics. European Child & Adolescent Psychiatry, 27(7), 909–919 [20] | obs & comp, retro, x-sect | (a) Center of Expertise on Gender Dysphoria (CEGD), Amsterdam, Netherlands; | 959 | 12–18 | (a) 46·0 | 2009–2013 | 4 |
| | | | | (b) Pediatric Gender Clinic, Ghent University Hospital, Belgium; | | | (b) 33·8 | | 4 |
| | | | | (c) Department of Child and Adolescent Psychiatry, University Hospital of Psychiatry Zurich, Switzerland; | | | (c) 35·7 | | 4 |
| | | | | (d) Gender Identity Development Service, Tavistock and Portman NHS Foundation Trust, London, UK | | | (d) 42·3 | | 4 |
| 13 | Multi | de Vries, et al. (2016). Poor peer relations predict parent- and self-reported behavioral and emotional problems of adolescents with gender dysphoria: a cross-national, cross-clinic comparative analysis. European Child and Adolescent Psychiatry, 25(6), 579–588 [42] | obs, comp, x-sect | 1. Center of Expertise on Gender Dysphoria (CEGD), Amsterdam | 139 | 13–18 | 56·8 | 1996–2008 | 1 |
| | | | | 2. Transgender Youth Clinic (TYC), The Hospital for Sick Children, Toronto, Canada | 177 | 13–18 | 53·1 | 1980–2010 | 4 |
| 14 | Netherlands | Cohen-Kettenis & Van Goozen (2002). Adolescents who are eligible for sex reassignment surgery: Parental reports of emotional and behavioural problems. Clinical Child Psychology and Psychiatry, 7(3), 412–422 [43] | obs, retro, x-sect | University Medical Centre, Utrecht (moved to VUmc / CEGD in 2002) | CBCL: 29  DISC: 21 | 11–18 | 37·9 | None given | 1 |

(*Continued*)

**Table 2.** (Continued)

| ID | Country | Reference | Design | Setting | N | Age (years) | Male natal sex (%) | Date range | GD status |
|---|---|---|---|---|---|---|---|---|---|
| 15 | Nether-lands | de Vries, et al. (2011a). Psychiatric comorbidity in gender dysphoric adolescents. Journal of child psychology and psychiatry, and allied disciplines, 52(11), 1195–1202 [44] | obs, prosp, x-sect | VU University Medical Center, Amsterdam (forerunner to CEGD) | 105 | 10·5–18·0 | 50·5 | Apr 2002-Dec 2009 | 1 |
| 16 | Nether-lands | de Vries, et al. (2011b). Puberty suppression in adolescents with gender identity disorder: A prospective follow-up study. Journal of Sexual Medicine, 8(8), 2276–2283 [45] | obs & comp, prosp, longit | VU University Medical Center, Amsterdam (forerunner to CEGD) | 70 | 11·1–17·0 | 47·1 | 2000–2008 | 1 |
| 17 | Nether-lands | de Vries, et al. (2014). Young adult psychological outcome after puberty suppression and gender reassignment. Pediatrics, 134(4), 696–704 [6] | obs, prosp, longit | Center of Expertise on Gender Dysphoria (CEGD), Amsterdam | 55 | 11·1–17·0 | 40 | 2004–2011 | 1 |
| 18 | Nether-lands | Klaver, et al. (2018). Early Hormonal Treatment Affects Body Composition and Body Shape in Young Transgender Adolescents. Journal of Sexual Medicine, 15(2), 251–260 [46] | obs, retro, longit, interv | VU University Medical Center, Amsterdam (forerunner to CEGD) | 192 | Mean at start puberty blocker NM 14.5±1.8 NF 15.3±2.0 | 37 | 1998–2014 | 1 |
| 19 | Nether-lands | Klaver, et al. (2020). Hormonal Treatment and Cardiovascular Risk Profile in Transgender Adolescents. Pediatrics, 145(3), 03 [47] | obs, retro, longit, interv | VU University Medical Center, Amsterdam (forerunner to CEGD) | 192 | Mean at start of puberty blocker NM 14.6±1.8 NF 15.2±2.0 | 37 | 1998–2015 | 1 |
| 20 | Nether-lands | Schagen, et al. (2012). Sibling sex ratio and birth order in early-onset gender dysphoric adolescents. Archives of Sexual Behavior, 41(3), 541–549 [48] | obs & comp, retro, x-sect | VU University Medical Center, Amsterdam (forerunner to CEGD) | 189 (control: 1789 age 7.6–14.0). | 9·0–17·9 | 49·7 | 1997–2008 | 1 |
| 21 | Nether-lands | van der Miesen, et al. (2018). Autistic Symptoms in Children and Adolescents with Gender Dysphoria. Journal of Autism and Developmental Disorders, 48(5), 1537–1548 [49] | obs & comp, prosp, x-sect | Center of Expertise on Gender Dysphoria (CEGD), Amsterdam | 490 (Comp grp 1: 2507 from Dutch schools. Comp grp 2: 196 with ASD dx Dutch psychiatry clinic. | Sample mean: 11·1 Comp grp 1 mean: 10·1 Comp grp 2 mean: 10·8 | 50·6 | Sample: 2005–2012 Comp grp 1: 1996–2000 Comp grp 2: 1996–2000 | 1 |
| 22 | Turkey | Akgul, et al. (2018). Autistic Traits and Executive Functions in Children and Adolescents With Gender Dysphoria. Journal of Sex & Marital Therapy, 44(7), 619–626 [50] | obs & comp, prosp, x-sect | Marmara University Pendik Education and Training Hospital's Child and Adolescent Psychiatry Clinic | GD: 25; control: 50 | 5–17 | 52 | Not given | 1 |
| 23 | UK | Costa, et al. (2015). Psychological Support, Puberty Suppression, and Psychosocial Functioning in Adolescents with Gender Dysphoria. Journal of Sexual Medicine, 12(11), 2206–2214 [24] | obs & comp, retro, longit, interv | Gender Identity Development Service, Tavistock & Portman, London | 201 (Control: 169 CAMHS Stockholm cases) | 12–17 | 37·8 | 2010–2014 | 1 |

(*Continued*)

**Table 2.** (Continued)

| ID | Country | Reference | Design | Setting | N | Age (years) | Male natal sex (%) | Date range | GD status |
|---|---|---|---|---|---|---|---|---|---|
| 24 | UK | Holt, V., et al. (2016). Young people with features of gender dysphoria: Demographics and associated difficulties. Clinical Child Psychology and Psychiatry, 21(1), 108–118 [26] | obs, retro, x-sect | Gender Identity Development Service, Tavistock & Portman, London | 218 | 5–17 (separate data on adols) | 37·2 | Jan-Dec 2012 | 2 |
| 25 | UK | Joseph, et al. (2019). The effect of GnRH analogue treatment on bone mineral density in young adolescents with gender dysphoria: findings from a large national cohort. Journal of pediatric endocrinology & metabolism: JPEM., 31 [51] | obs, retro, longit, interv | Gender Identity Development Service, Tavistock & UCLH Early Intervention programme @ national endocrine clinic, London | 70 | 12–14 | 44·3 | 2011–2016 | 2 |
| 26 | UK | Matthews, et al. (2019). Gender Dysphoria in looked-after and adopted young people in a gender identity development service. Clinical Child Psychology & Psychiatry, 24(1), 112–128 [23] | obs, retro, x-sect | Gender Identity Development Service, Tavistock & Portman, London | 185 | 3–17 | 52·4 | 2009–2011 | 1 |
| 27 | UK | Russell, et al. (2020). A Longitudinal Study of Features Associated with Autism Spectrum in Clinic Referred, Gender Diverse Adolescents Accessing Puberty Suppression Treatment. Journal of Autism and Developmental Disorders [52] | obs, retro, x-sect | Gender Identity Service, Tavistock & Portman, London | 95 | 9·9–15·9 (mean 13·6 ±0·11) | 40 | Not given | 2 |
| 28 | UK | Skagerberg, et al. (2013). Internalizing and externalizing behaviors in a group of young people with gender dysphoria. International Journal of Transgenderism, 14(3), 105–112 [53] | obs, x-sect | Gender Identity Development Service (GIDS), Tavistock & Portman, London | 141 | 12–18 | 40 | None given | 1 |
| 29 | USA | Becerra-Culqui, et al. (2018). Mental health of transgender and gender nonconforming youth compared with their peers. Pediatrics, 141 (5) (e20173845) [28] | obs & comp, retro, x-sect | California and Georgia, US, Kaiser-Permanente records | 1082 (control: 21,317 cis gender matched (≥7 per index case)) | 10–17 | 39·5 | 1/1/06–31/12/14 | 3 |
| 30 | USA | Chen, et al. (2016). Characteristics of Referrals for Gender Dysphoria over a 13-Year Period. Journal of Adolescent Health, 58(3), 369–371 [21] | obs, retro, x-sect | paediatric endocrinology clinic, Indiana | 38 | Mean age 14·4±3·2 | 42·1 | 1/1/02–1/4/15 | 3 |
| 31 | USA | Edwards-Leeper, et al. (2017). Psychological profile of the first sample of transgender youth presenting for medical intervention in a US pediatric gender center. Psychology of Sexual Orientation and Gender Diversity, 4(3), 374–382 [54] | obs, retro, x-sect | The Gender Management Service (GeMS) program at Boston Children's Hospital | 56 | 8·9–17·9 | 53·6 | 2007–2011 | 4 |

(*Continued*)

**Table 2.** (Continued)

| ID | Country | Reference | Design | Setting | N | Age (years) | Male natal sex (%) | Date range | GD status |
|---|---|---|---|---|---|---|---|---|---|
| 32 | USA | Kuper, et al. (2019a). Exploring the gender development histories of children and adolescents presenting for gender affirming medical care. Clinical Practice in Pediatric Psychology, 7(3), 217–228 [55] | obs, retro, x-sect | 'gender affirming service', University of Texas Southwestern Medical Center* | 224 | 6–17 | 39·4 | 2014–2018 | 1 |
| 33 | USA | Kuper, et al. (2019b). Baseline Mental Health and Psychosocial Functioning of Transgender Adolescents Seeking Gender-Affirming Hormone Therapy. Journal of developmental and behavioral pediatrics: JDBP., 03 [56] | obs, prosp, x-sect | Gender Education and Care Interdisciplinary Support program, Texas | 149 | 12–18 | 35·6 | 2014–2017 | 1 |
| 34 | USA | Kuper, et al. (2020). Body Dissatisfaction and Mental Health Outcomes of Youth on Gender-Affirming Hormone Therapy. Pediatrics, 145(4), 04 [57] | obs, prosp, longit, interv | 'a multidisciplinary program in Dallas, Texas' | 148 | 9–18 | 37 | Initially assessed 2014–2018 | 1 |
| 35 | USA | Lopez, et al. (2018). Trends in the use of puberty blockers among transgender children in the United States. Journal of Pediatric Endocrinology and Metabolism, 31(6), 665–670 [58] | obs, retro, x-sect | US Pediatric Health and Information System (PHIS) database | 92 | 8·8–17·8[a] | 46·3 | 2010-2015[b] | 3 |
| 36 | USA | Moyer, et al. (2019). Using the PHQ-9 and GAD-7 to screen for acute distress in transgender youth: findings from a pediatric endocrinology clinic. Journal of Pediatric Endocrinology & Metabolism, 32(1), 71–74 [59] | obs, retro, x-sect | 'a pediatric endocrinology clinic', Portland, Oregon | 194 | 11–18 | 24·2 | None given | 2 |
| 37 | USA | Nahata, et al. (2017). Mental Health Concerns and Insurance Denials Among Transgender Adolescents. LGBT Health, 4(3), 188–193 [60] | obs, retro, x-sect | 'gender program' at an 'urban, Mid-western, pediatric academic center', Nationwide Children's Hospital, Columbus, Ohio* | 79 | 9–18 | 35·4 | 2014–2016 | 3 |
| 38 | USA | Olson-Kennedy, et al. (2019). Impact of Early Medical Treatment for Transgender Youth: Protocol for the Longitudinal, Observational Trans Youth Care Study. JMIR Research Protocols, 8(7), e14434 [61] | obs, prosp, longit, interv | Children's Hospital Los Angeles/University of Southern California, Boston Children's Hospital/Harvard University, Lurie Children's Hospital of Chicago/Northwestern University, Benioff Children's Hospital/University of California San Francisco | Blocker cohort: 90 | 8–16 | 51 | July 2016-Sept 2018 | 1 |

Key:

* = author's affiliation. No specific setting given     obs = observational     comp = comparative

$ = authors report sample overlap     prosp = prospective     retro = retrospective

* = inferred–not overtlay stated     longit = longitudinal     x-sect = cross-sectional

€ = age of first evidence of transgender status in medical records     interv = intervention study

a = Max age of whole sample 18.8. Only data on those under 18 years used in this review.

B = 2016 data not used in this review as included participants over 18 years

GD status codes: 1) clinical diagnosis using DSM-III / IV / IV-TR / 5; 2) active patients within clinic; 3) data mined using ICD 9 / 10 codes and/or relevant keywords; 4) Under assessment at clinic–beyond referral stage

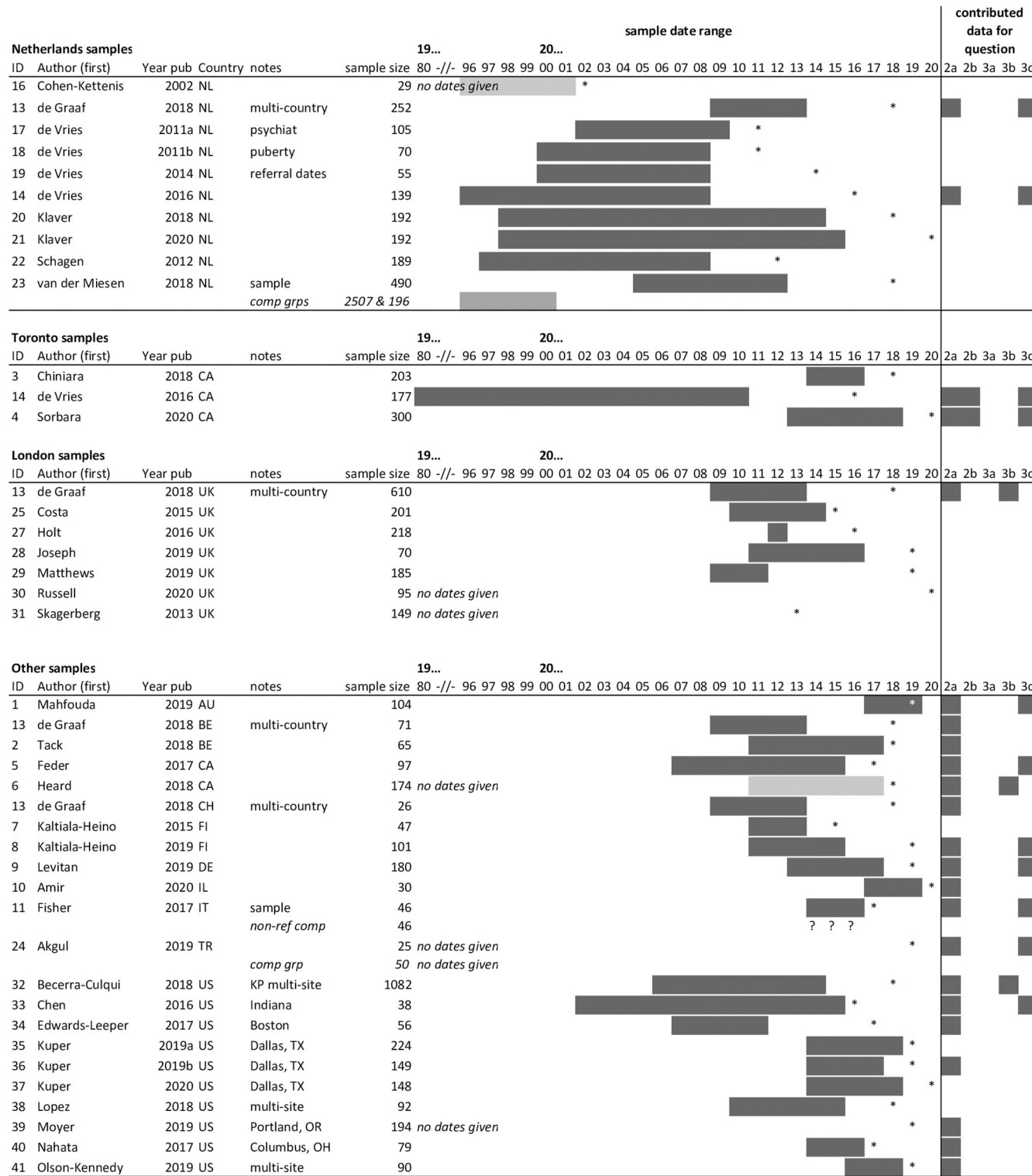

**Fig 2. Overlap between included samples.** Key -//-: dates truncated; *: year of publication; NL: Netherlands; CA: Canada; UK: United Kingdom; AU: Australia; BE: Belgium; FI: Finland; DE: Germany; IL: Israel; IT: Italy; CH: Switzerland; TR: Turkey; US: United States.

interested in adolescents seeking and considered eligible for intervention. Given that Amsterdam and London centres are the only specialist centres in their respective countries, both of which have state-funded health systems, it would not be unreasonable to use their data as a likely indication of incidence / prevalence. The figures reported in de Graaf et al. [20] give the largest and most recent sample from these 2 locations (252 and 610 respectively), but these are sub-samples of the clinic population for whom data were available, and they do not comment on prevalence as a proportion of the population. It is possible to say there has been an increase in adolescents presenting for treatment in recent years. For example, Chen et al. [21] report the majority of their sample (73·6%) presented for treatment in the final two years of a 13 year period (2002–2015).

**2. What are the proportions of natal males / females with GD in adolescence and has this changed over time?.** All included studies featured data on participants' natal sex, usually at the time of first being assessed by a specialist gender clinic. A simple pooling of proportions from all the papers indicates 36% were natal males and 64% natal females. Restricting our analysis only to those studies we could be certain had distinct samples (and aiming to select the largest / widest date range within, see Fig 2), the proportions remained similar at 37% natal male and 63% natal female.

One paper addressed the question of a recent shift in natal sex ratio directly. Chiniara et al. [22] conducted a within-sample analysis of their 2014–2016 referred participants in Toronto and found no change in that short time period. They also compared the natal sex ratio in their sample to those previously published and found a shift in more recent years (1:3 favouring natal females vs 0·8–0·9:1 in earlier studies).

Although only a few papers in our sample addressed the question directly, we used pooled data to explore whether there is evidence of a shift in recent years to more natal boys or girls seeking assessment / treatment. Papers were grouped into three categories according to the date range that samples were assessed: pre-2000; 2001–2010; 2011 onward. This was challenging as most studies were retrospective chart reviews covering wide date ranges from the late 1980s to beyond 2010. However, it is possible to say that the proportion of natal males is slightly lower (30%) in those papers featuring participants assessed only from 2011 onwards (ten papers). These data are summarised in Table 2.

**3. What is the pattern of age at (a) onset (b) referral and (c) assessment?.** Only one of the included papers focused specifically on age of onset: Matthews et al. [23] reported a mean age of onset of 6·80 years (SD 3·9) (range 1–15) among 168 referrals to the London GIDS. Six papers reported explicitly on age of referral (Costa et al., 2015 [24]; de Graaf et al. 2018 [20]; Heard et al. 2018 [25]; Holt et al. [26], 2016; Mahfouda et al. [27]; Matthews et al. 2019 [23]) giving a pooled mean age of referral as 13·2 years (SD 0·9) (data from Costa et al., 2015 [24], excluded due to overlap with de Graaf et al., 2018 [20]). One paper (Becerra-Culqui et al., 2018 [28]) used medical records and took participants' age from 'the first evidence of transgender and/or gender nonconforming status' based on the presence of certain keywords in medical notes. Some papers (e.g., Chen et al., 2016 [21]) were not explicit about whether they were reporting age at referral or assessment, but usually it could be inferred that the reported age was at assessment.

Age was most usually reported at point of assessment or intervention (see Table 3). Not all papers reported full age data: a mean and standard deviation was usually given, but not always a range. The pooled mean age of assessment was 15·1 years (SD 1·0) and the range (from fewer papers) was 6·0–18·0 years.

**Table 3. Detailed age data used in analysis.**

| ID | reference | | date | Age at referral (years) | | | |
|---|---|---|---|---|---|---|---|
| | | | | Mean | SD | Range | N |
| 25 | Costa, et al. | | 2015 | 15·52 | 1·41 | 12–17 | 201 |
| 13 | De Graaf et al. | - sample 1 Netherlands | 2018 | 14·3 | 2·18 | - | 252 |
| | | - sample 2 Belgium | | 14·34 | 1·65 | - | 71 |
| | | - sample 3 Switzerland | | 15·38 | 1·2 | - | 26 |
| | | - sample 4 UK | | 15·54 | 1·58 | - | 610 |
| 5 | Heard, et al. | | 2018 | 13·9 | - | 4·7–17·8 | 174 |
| 27 | Holt et al. | | 2016 | 14 | 3·08 | 5·0–17 | 218 |
| 1 | Mahfouda et al. | | 2019 | 14·62 | 1.72 | - | 104 |
| 29 | Matthews, et al. | | 2019 | 13·7 | 3·35 | 3–17 | 185 |
| | | | | Age at assessment (years) | | | |
| 24 | Akgul, et al. | | 2018 | 11.6 | 4.2 | 6.0–18.0 | 25 |
| 10 | Amir et al. | | 2020 | - | - | <18 | 30 |
| 32 | Chen, et al. | | 2016 | 14.4 | 3.2 | | 38 |
| 3 | Chiniara, et al. | - sample 1 (AFAB) | 2018 | 16.3 | 1.6 | 12.0–18.0* | 156 |
| | | - sample 2 (AMAB) | | 16.1 | 1.7 | 12.0–18.0* | 47 |
| 16 | Cohen-Kettenis & Van Goozen | - sample 1 (MF, CBCL)[a] | 2002 | 15.1 | 2.0 | 11.0–17.0 | 11 |
| | | - sample 2 (FM, CBCL) | | 15.2 | 2.1 | 11.0–18.0 | 18 |
| 17 | de Vries, et al. | | 2011 | 14.6 | 2.2 | 10.5–18.0 | 105 |
| 18 | de Vries, et al. | | 2011 | 13.7 | 1.9 | 11.1–17.0 | 70 |
| | de Vries, et al. | | 2014 | 13.6 | 1.9 | 11.1–17.0 | 55 |
| | de Vries, et al. | - sample 1 (Netherlands) | 2016 | 15.7 | 1.5 | 13.0–18.0 | 139 |
| | | - sample 2 (Canada) | | 15.9 | 1.3 | 13.0–18.0 | 177 |
| 33 | Edwards-Leeper, et al. | | 2017 | - | - | 8.9–17.9 | 56 |
| 4 | Feder, et al. | | 2017 | 15.7 | 1.4 | 12.0–18.0* | 97 |
| 11 | Fisher, et al. | | 2017 | 16.4 | 1.3 | - | 46 |
| 28 | Joseph, et al.[b] | - sample 1 (transgirls) | 2019 | 13.0 | 1.1 | - | 10 |
| | | - sample 2 (transboys) | | 12.9 | 3.0 | - | 21 |
| 7 | Kaltiala-Heino et al. | - sample 1 NM | 2015 | 16.0 | 0.6 | - | 6 |
| | | - sample 2 NF | | 16.7 | 1.1 | - | 41 |
| 8 | Kaltiala-Heino et al. | | 2019 | 16.9 | 0.9 | - | 107 |
| 34 | Kuper, et al. | | 2019a | 14.7 | 2.5 | 6.0–17.0 | 224 |
| 35 | Kuper, et al. | | 2019b | 15.3 | 1.5 | 12.0–18.0 | 149 |
| 36 | Kuper, et al. | | 2020 | 15.4 | 2.0 | 9.0–18.0 | 148 |
| 9 | Levitan, et al. | | 2019 | 15.5 | 1.4 | - | 180 |
| 1 | Mahfouda, et al. | | 2019 | 15.49 | 1.67 | - | 104 |
| 38 | Moyer, et al. | | 2019 | 15.6 | 1.8 | 11.0–18.0 | 194 |
| 39 | Nahata, et al.[c] | | 2017 | 15.0 | - | 9.0–18.0 | 79 |
| 40 | Olson-Kennedy, et al. | | 2019 | 11.0 | 1.5 | 8.0–16.0 | 90 |
| 30 | Russell, et al. | | 2020 | 13.6 | 0.1 | 9.9–15.9 | 122 |
| 21 | Schagen, et al. | - sample 1 (GID boys) | 2012 | 13.4 | - | 9.0–17.7 | 74 |
| | | - sample 2 (GID girls) | | 13.6 | - | 9.1–17.9 | 85 |
| | | - sample 3 (boys GID indicators) | | 12.6 | - | 9.2–17.8 | 20 |
| | | - sample 4 (girls GID indicators) | | 12.8 | - | 9.0–17.0 | 10 |
| | Skagerberg, et al. | | 2013 | 15.1 | 1.7 | 12.0–18.0 | 141 |
| 6 | Sorbara, et al. | | 2020 | 15.4 | | 14.2–16.4[d] | 300 |

(*Continued*)

**Table 3.** (*Continued*)

| ID | reference | date | Age at referral (years) | | | |
|---|---|---|---|---|---|---|
| | | | Mean | SD | Range | N |
| 23 | van der Miesen, et al. | 2018 | 11.1 | 3.73 | - | 490 |

Key:

* = eligibility range, not necessarily observed range

a—described CBCL sample as the larger of the two included. Overlap between samples means not ideal to describe both

b—described only 31 subcohort who had all 3 scans (BL, FU1, FU2)

c—age data not included in pooled means as only medians available

d–Inter-quartile range

grey shading = excluded from pooled analysis due to overlapping samples

NB: terminology such as 'transboys' or 'GID girls' is taken verbatim from the paper.

## Quality assessment

The CCAT quality ratings ranged from 45% to 96%, with a mean of 78%. Most papers achieved an overall rating of 4 (good) or 5 (very good), with strengths and weaknesses within certain discrete categories; most papers achieved good ratings in the 'preliminaries' and 'introduction' categories, whereas the 'ethics' and 'discussion' categories were most likely to include lower ratings: 17 and 16 papers respectively achieved ratings below 4. In total, only one paper was rated as 3 (moderate quality): Cohen-Kettenis & Van Goozen (2002) obtained low ratings across most categories, due to unclear sampling and diagnostic information, lack of information to permit replication, and conclusions which are not supported by the findings. Of the remainder, 16 were rated as high quality (4), and 21 as very high quality (5; see Table 4). There was no relationship between the year of publication and quality rating ($r = 0.2$).

## Discussion

This systematic review synthesises the current evidence regarding the age and natal sex of adolescents presenting to specialist services and assessed as having gender dysphoria (GD). Based on 38 papers meeting inclusion criteria, there is evidence of an increase in frequency of presentation to services since 2011, and of a shift in the natal sex of referred cases: those assigned female at birth are now in the majority. Within these samples the average age of referral was 13 years, and the average age of assessment was 15 years. This review is the first of its kind to focus on adolescent samples where diagnostic criteria for GD were met, or significant GD features were clinically verified.

Although other good quality review papers have been published [12, 17, 29–31], they have tended not to apply a systematic review methodology or have taken a broader scope in their inclusion criteria. We believe this is the first systematic review focused only on adolescents aged under 18 years and on clinically-verified samples taking into account likely study overlap.

Due to a lack of population-based research including cases of clinically-verified GD, this review was unable to report overall prevalence of GD (although it would not be unreasonable to use the Amsterdam and London data to make a good estimate). At present, the only means of estimating prevalence is to use population-based survey data, which carries risk of respondent bias (and such papers were excluded from our sample). Some studies used administrative records to ascertain samples of adolescents with GD. The reliability of this method is dependent on GD being accurately recorded, and on administrative data systems having universal

**Table 4. Quality ratings using Crowe Critical Appraisal Tool (CCAT).**

| ID | Country | Reference | Average CCAT rating | | | | | | | | | | |
|---|---|---|---|---|---|---|---|---|---|---|---|---|---|
| | | | 1 | 2 | 3 | 4 | 5 | 6 | 7 | 8 | | | |
| | | | Preliminaries | Introduction | Design | Sampling | Data collection | Ethical matters | Results | Discussion | Total | % | Overall level |
| 1 | Australia | Mahfouda, et al. (2019). | 4 | 4·5 | 3·5 | 4 | 3·5 | 5 | 4 | 4 | 32·5 | 81 | 5 |
| 2 | Belgium | Tack, et al. (2018). | 4·5 | 4 | 3·5 | 3·5 | 4·5 | 4·5 | 4 | 4 | 32·5 | 81 | 5 |
| 3 | Canada | Chiniara, et al. (2018). | 5 | 5 | 4·5 | 4·5 | 4 | 3·5 | 3·5 | 4 | 34 | 85 | 5 |
| 4 | Canada | Feder, et al. (2017). | 4·5 | 5 | 4·5 | 5 | 4 | 2 | 3·5 | 3·5 | 32 | 80 | 4 |
| 5 | Canada | Heard, et al. (2018). | 4·5 | 5 | 4 | 4 | 3·5 | 2·5 | 3·5 | 4 | 31 | 78 | 4 |
| 6 | Canada | Sorbara, et al. (2020). | 5 | 4 | 4 | 4 | 4 | 2 | 4 | 4·5 | 31·5 | 79 | 4 |
| 7 | Finland | Kaltiala-Heino, et al. (2015). | 4·5 | 5 | 5 | 5 | 4·5 | 4·5 | 4·5 | 5 | 38 | 95 | 5 |
| 8 | Finland | Kaltiala-Heino, et al. (2019). | 5 | 5 | 4·5 | 5 | 4·5 | 5 | 5 | 4·5 | 38·5 | 96 | 5 |
| 9 | Germany | Levitan, et al. (2019). | 5 | 5 | 5 | 5 | 4·5 | 4·5 | 5 | 4·5 | 38·5 | 96 | 5 |
| 10 | Israel | Amir, et al (2020). | 4 | 4·5 | 4·5 | 4 | 4 | 3·5 | 4 | 4·5 | 33 | 83 | 5 |
| 11 | Italy | Fisher, et al. (2017). | 4 | 4·5 | 4 | 3·5 | 4 | 4 | 4 | 4·5 | 32·5 | 81 | 5 |
| 12 | Multi | de Graaf, et al. (2018a). | 5 | 4·5 | 4·5 | 4·5 | 4·5 | 4·5 | 4 | 4 | 35·5 | 89 | 5 |
| 13 | Multi | De Vries, et al. (2016) | 4·5 | 4·5 | 4 | 3·5 | 3·5 | 2·5 | 4 | 3·5 | 30 | 75 | 4 |
| 14 | N/lands | Cohen-Kettenis & van Goozen (2002). | 2·5 | 2·5 | 3 | 2 | 2·5 | 0 | 3 | 2·5 | 18 | 45 | 3 |
| 15 | N/lands | de Vries, et al. (2011a). | 5 | 5 | 3·5 | 4·5 | 5 | 4·5 | 4·5 | 3·5 | 35·5 | 89 | 5 |
| 16 | N/lands | de Vries, et al. (2011b). | 5 | 4 | 2·5 | 3·5 | 4·5 | 4 | 4 | 3 | 30·5 | 76 | 4 |
| 17 | N/lands | De Vries, et al. (2014) | 4 | 5 | 3·5 | 4 | 3·5 | 4 | 4·5 | 3·5 | 32 | 80 | 4 |
| 18 | N/lands | Klaver, et al. (2018). | 4·5 | 4·5 | 5 | 5 | 5 | 5 | 4·5 | 4·5 | 38 | 95 | 5 |
| 19 | N/lands | Klaver, et al. (2020). | 4 | 4·5 | 4·5 | 4 | 4 | 4·5 | 4 | 4·5 | 34 | 85 | 5 |
| 20 | N/lands | Schagen, et al. (2012). | 4·5 | 5 | 5 | 4·5 | 4·5 | 3 | 5 | 4 | 35·5 | 89 | 5 |
| 21 | N/lands | van der Miesen, et al. (2018). | 5 | 5 | 4·5 | 4·5 | 4 | 5 | 4·5 | 4 | 36·5 | 91 | 5 |
| 22 | Turkey | Akgul, et al. (2018). | 4 | 4·5 | 4·5 | 4·5 | 4 | 2·5 | 4 | 4 | 32 | 80 | 4 |
| 23 | UK | Costa, et al. (2015). | 4·5 | 4·5 | 4 | 4·5 | 4 | 3·5 | 4 | 4·5 | 33·5 | 84 | 5 |
| 24 | UK | Holt, V., et al. (2016). | 4 | 4 | 3·5 | 5 | 3·5 | 3·5 | 4 | 3·5 | 31 | 78 | 4 |
| 25 | UK | Joseph, et al. (2019). | 4 | 4 | 3·5 | 3·5 | 4 | 4 | 3·5 | 3 | 29·5 | 74 | 4 |
| 26 | UK | Matthews, et al. (2019). | 4 | 5 | 4 | 4·5 | 4 | 5 | 3·5 | 4 | 34 | 85 | 5 |
| 27 | UK | Russell, et al. (2020). | 5 | 5 | 3·5 | 3·5 | 4 | 2·5 | 3·5 | 3·5 | 30·5 | 76 | 4 |
| 28 | UK | Skagerberg, et al. (2013) | 4·5 | 5 | 4 | 3·5 | 4 | 1 | 4 | 4·5 | 30·5 | 76 | 4 |
| 29 | USA | Becerra-Culqui, et al. (2018). | 4·5 | 4·5 | 4·5 | 4·5 | 4·5 | 4·5 | 4·5 | 4 | 35·5 | 89 | 5 |
| 30 | USA | Chen, et al. (2016). | 4 | 3·5 | 4 | 4 | 4 | 2·5 | 3·5 | 3 | 28·5 | 71 | 4 |
| 31 | USA | Edwards-Leeper, et al. (2017). | 4·5 | 5 | 4·5 | 3·5 | 5 | 3 | 4 | 4·5 | 34 | 85 | 5 |
| 32 | USA | Kuper, et al. (2019a). | 3·5 | 4 | 3·5 | 4 | 4 | 3·5 | 4 | 3·5 | 30 | 75 | 4 |
| 33 | USA | Kuper, et al. (2019b). | 4·5 | 4·5 | 4 | 4 | 3·5 | 4 | 4 | 3·5 | 32 | 80 | 4 |
| 34 | USA | Kuper, et al. (2020). | 5 | 5 | 4 | 4·5 | 5 | 3 | 4 | 3·5 | 34 | 85 | 5 |
| 35 | USA | Lopez, et al. (2018). | 4 | 4 | 4·5 | 4·5 | 4 | 4 | 3·5 | 3 | 31·5 | 79 | 4 |
| 36 | USA | Moyer, et al. (2019). | 4·5 | 4 | 4·5 | 4·5 | 4 | 4 | 4 | 3·5 | 33 | 83 | 5 |

*(Continued)*

**Table 4.** (Continued)

| ID | Country | Reference | Average CCAT rating | | | | | | | | Total | % | Overall level |
|----|---------|-----------|------|------|------|------|------|------|------|------|-------|---|---------------|
| | | | 1 | 2 | 3 | 4 | 5 | 6 | 7 | 8 | | | |
| | | | Preliminaries | Introduction | Design | Sampling | Data collection | Ethical matters | Results | Discussion | | | |
| 37 | USA | Nahata, et al. (2017). | 4 | 4 | 4 | 4 | 4 | 4 | 4 | 3 | 31 | 78 | 4 |
| 38 | USA | Olson-Kennedy, et al. (2019). | 4·5 | 5 | 3·5 | 5 | 4·5 | 4 | 4·5 | 4 | 35 | 88 | 5 |

coverage. These criteria could not be met by the samples included in the present review and should remain a focus for development in future research.

This review confirmed that the increase in referrals and the shift in sex-ratio that has been observed more widely in survey data and referred populations is also present in clinically-verified samples. We were able to report data from relatively few papers, however. Given the size of the literature, it would be useful if more studies clearly reported or clearly differentiated samples according to the stage of identification / referral / assessment participants had reached. It is clear that many of the samples reported in this review began as much larger samples with significant attrition before completing assessment and / or intervention. Not all papers fully reported either attrition figures or reasons for drop-out. Most papers were cross-sectional descriptions of cohorts of clinic patients with no longer term follow-up or comparison with control or normative samples, which limits the external generalisability of the findings.

We were unable to report on age of onset of GD as this was rarely reported. Where it was reported, it was based on patient / parent recall and difficult to verify clinically (e.g., Matthews et al. [23] report a lower age of onset of 1 year). Age of referral also proved difficult to report on as most samples that could be classed as clinically-verified were further along the line in their clinical journey. Whilst some papers reported age at referral, it was not always clear whether 'referral' meant the age at which a young person first sought help and was referred to a specialist service, or the age at which they had their first contact with that service. Given that most treatment takes place at national specialist centres, waiting lists may mean that age of first specialist contact may be two or three years beyond age at first referral. This will vary according to the economic basis of national health systems.

Age at assessment data reported in this review suggests that young people may not be receiving specialist support when they need it. The Dutch model, developed by the Center of Expertise on GD in the Netherlands where GD interventions have been pioneered, considers age 12 years to be the lower threshold for puberty suppression treatment, and age 16 as the threshold for cross-sex hormone treatment [6]. Participants in our samples had an average age of 15 years at the time of assessment, and so are likely to have already undergone significant pubertal changes associated with their natal sex. It may be that young people are first presenting to services once pubertal changes have begun and GD has become established, but delays between first presentation to services and being seen in a specialist service may also be important. It is possible that critical windows of opportunity for intervention are being missed. Our third review paper will focus on the data regarding age at intervention and related outcomes in more detail. More transparent reporting of age data and consistent use of terminology would allow us to better understand the clinical landscape and could inform rational service development.

## Strengths and limitations

This review has strength in the broad search strategy and thorough hand screening process applied. There are methodological limitations which need to be considered. The application of strict inclusion criteria in a rapidly growing field, with new findings emerging on an almost daily basis, means it is impossible to be completely up to date. For example, Zucker and Aitken (2019) [32] recently confirmed the shift in sex ratio of transgender adolescents in a large meta-analysis, but this was not included as it is, at present, a conference proceeding and not a peer-reviewed paper. The broad initial search criteria led to the need for some narrowing of criteria following initial screening (but prior to full-text screening). The addition of parameters regarding type of publication, upper age of participants, and the clinical verification of GD naturally narrowed the pool of papers and therefore may have meant papers with important findings have been excluded (for example, if a paper included an upper age limit of 21 even though the majority were younger than 18). We recorded all papers that only narrowly missed inclusion on the age criterion (Table 5), but this will naturally have led to important papers in the field being missed from the review. For example, the paper by Aitken et al. (2015) had to be excluded as in Sample 1 (Canadian) was the eldest was 19 years old, and in Sample 2 (Netherlands) diagnostic status was unclear. Alberse et al. (2019) [33] had an upper age range of 18.03, requiring its exclusion. These were otherwise good papers that we would like to have been able to include. However, it was important that we consistently apply our *a priori* criteria at every stage of screening, even when it meant that important papers only very narrowly missing inclusion had to be excluded. To apply flexibility only at the final screen presented a risk to the integrity of screening at earlier stages. We do not wish to give the impression that the importance of these papers has not been considered. There are several non-systematic reviews available that will include these high-profile papers. Our objective was to take in the totality of the literature and then examine the state of the evidence once strict criteria were applied.

Due to the scale of the overall review, no formal hand searching was included, although we did check whether any relevant texts cited in our review of papers had been included in our sample, and they had (either excluded at an earlier stage or included in the final sample). We opted to use a quality assessment tool for studies of diverse designs (CCAT). This allowed all papers to be rated using the same system, but also involved reviewers having to make subjective ratings rather than apply a strictly quantifiable checklist. This may have led to issues with quality, such as over-statement of the significance of findings, not being sufficiently prominent. The quality of the literature is mixed, and we were unable to clearly answer all the research questions. Although this presents a limitation to this review, it also constitutes an important finding in and of itself.

Although we were able to include 38 papers from a range of countries in this review, almost half arose from two well-established treatment centres: those in Amsterdam and London. The Amsterdam team has led the way in developing assessment and treatment protocols for GD and provides a wealth of data over a long period (since 1996 within the included papers), and the London GIDS is a hub for the whole of the UK now dealing with hundreds of referrals per year. This presents the advantage of being able to observe the adolescent GD population over a long period of time, assessed using the same or similar tools, and within a relatively stable social context. It is not clear, however, what proportion of young people experiencing GD have access to these national specialist centres and how many may be accessing private facilities or self-medicating with hormones obtained via other routes: we do not know how representative these samples are. Another disadvantage is that most of the papers included in this review are likely to include data from the same samples of participants, also limiting generalisability. The overlap between samples was rarely overtly stated, and there is a risk that readers may add

**Table 5. Papers excluded at second full text screen (i.e., closely missed meeting inclusion criteria).**

| Reference | Date | Location & setting | Reason for exclusion | Notes |
|---|---|---|---|---|
| Achille, C., Taggart, T., Eaton, N. R., Osipoff, J., Tafuri, K., Lane, A., & Wilson, T. A. (2020). Longitudinal impact of gender-affirming endocrine intervention on the mental health and well-being of transgender youths: preliminary results. International Journal of Pediatric Endocrinology, 2020, 8. | 2020 | USA<br>New York, Stoney Brook Children's Hospital | 1<br>Mean age 16.2±2.2 | 66% NF<br>MH improved with endocrine intervention<br>Small sample (n = 50) |
| Aitken, et al. (2015). Evidence for an altered sex ratio in clinic-referred adolescents with gender dysphoria. Journal of Sexual Medicine, 12(3), 756–763. | 2015 | 1) CANADA<br>Gender Identity Service, Child, Youth and Family Services (CYFS), Toronto<br>2) NETHERLANDS<br>Center of Expertise on Gender Dysphoria (CEGD) | 1 (N/LANDS), 2 (CANADA) | 1) unclear if GD clinically verified<br>2) Eldest participant 19 years Useful information in change of gender in those presenting to services over time. |
| Alastanos, J. N., & Mullen, S. (2017). Psychiatric admission in adolescent transgender patients: A case series. The Mental Health Clinician, 7(4), 172–175. | 2017 | USA<br>'an inpatient psychiatry unit' | 3<br>Selected case series | All 5 participants were psychiatric inpatients within a 5 week period |
| Alberse, et al. (2019). Self-perception of transgender clinic referred gender diverse children and adolescents. Clinical Child Psychology and Psychiatry, 24(2), 388–401. | 2019 | NETHERLANDS<br>Center of Expertise on Gender Dysphoria (CEGD), Amsterdam | 1<br>Max age 18.03 | Poor self-perception common. NF perceive themselves more positively in general. |
| Alexander, G. M., & Peterson, B. S. (2004). Testing the prenatal hormone hypothesis of tic-related disorders: gender identity and gender role behavior. Development & Psychopathology, 16(2), 407–420. | 2004 | USA<br>Child Study Center of Yale University in New Haven, CT | 6 | Participants receiving treatment for Tourette Syndrome, not presenting for GD |
| Amir, H., Oren, A., Klochendler Frishman, E., Sapir, O., Shufaro, Y., Segev Becker, A., . . . Ben-Haroush, A. (2020). Oocyte retrieval outcomes among adolescent transgender males. Journal of Assisted Reproduction & Genetics, 37(7), 1737–1744. | 2020 | ISRAEL<br>IVF Unit, Fertility Institute in Tel Aviv Sourasky Medical Center and IVF and the Infertility Unit, Helen Schneider Hospital for Women, Rabin Medical Center | 3 | Sample was 11 NF presenting specifically for fertility preservation |
| Anzani, A., Panfilis, C., Scandurra, C., & Prunas, A. (2020). Personality Disorders and Personality Profiles in a Sample of Transgender Individuals Requesting Gender-Affirming Treatments. International Journal of Environmental Research & Public Health, 17(5), 27. | 2020 | ITALY<br>gender clinic at Niguarda Ca' Granda Hospital, Milan | 3 | Very small sub-sample (n = 4) in adolescent age range |
| Arnoldussen, et al. (2019). Re-evaluation of the Dutch approach: are recently referred transgender youth different compared to earlier referrals? European Child and Adolescent Psychiatry. | 2019 | NETHERLANDS<br>Center of Expertise on Gender Dysphoria (CEGD), Amsterdam | 1<br>Max age 18.08 | From 2000–2016, sharp increase in cases from 2012 to 2016; sharp uptick in NF relative to NM since 2013 |
| Avila, J. T., Golden, N. H., & Aye, T. (2019). Eating Disorder Screening in Transgender Youth. Journal of Adolescent Health, 65(6), 815–817. | 2019 | USA<br>'an academic multidisciplinary gender clinic'<br>Stanford University School of Medicine* | 1, 2<br>No distinct data on adolescents in sample. | Most (63%) disclosed weight manipulation for gender-affirming purposes, including<br>11% of NF for menstrual suppression. |

*(Continued)*

**Table 5.** (Continued)

| Reference | Date | Location & setting | Reason for exclusion | Notes |
|---|---|---|---|---|
| Barnard, E. P., Dhar, C. P., Rothenberg, S. S., Menke, M. N., Witchel, S. F., Montano, G. T., . . . Valli-Pulaski, H. (2019). Fertility Preservation Outcomes in Adolescent and Young Adult Feminizing Transgender Patients. Pediatrics, 144(3). | 2019 | USA<br>Magee-Womens Research Institute, Pittsburgh* | 1, 3 | Very small subsample (n = 11); only 4 in adolescence at time of assessment (consultation). |
| Becerra-Fernández, A., Rodríguez-Molina, J., Ly-Pen, D., Asenjo-Araque, N., Lucio-Pérez, M., Cuchí-Alfaro, M., . . . Aguilar-Vilas, M. V. (2017). Prevalence, Incidence, and Sex Ratio of Transsexualism in the Autonomous Region of Madrid (Spain) According to Healthcare Demand. Archives of Sexual Behavior, 46(5), 1307–1312. doi:10.1007/s10508-017-0955-z | 2017 | SPAIN<br>Gender Identity Unit, Madrid | 1<br>No separate data on adolescents (includes ≥18 yrs) | Higher prevalence rate than other countries: attributed to easily accessible services and positive social and legal climate in Spain. |
| Bechard, M., VanderLaan, D. P., Wood, H., Wasserman, L., & Zucker, K. J. (2017). Psychosocial and Psychological Vulnerability in Adolescents with Gender Dysphoria: A "Proof of Principle" Study. Journal of Sex & Marital Therapy, 43(7), 678–688. | 2017 | CANADA<br>Gender Identity Service, CYFS, Toronto | 1<br>No separate data on adolescents (includes ≥18 yrs) | Mean of 5.56/13 'psychological vulnerability factors' amongst sample. |
| Becker, I., Auer, M., Barkmann, C., Fuss, J., Moller, B., Nieder, T. O., . . . Richter-Appelt, H. (2018). A Cross-Sectional Multicenter Study of Multidimensional Body Image in Adolescents and Adults with Gender Dysphoria Before and After Transition-Related Medical Interventions. Archives of Sexual Behavior, 47(8), 2335–2347. | 2018 | GERMANY<br>Department of Child and Adolescent Psychiatry, Psychotherapy, and Psycho-Somatics, University Medical Centre, Hamburg | 1<br>No separate data on adolescents (includes ≥18 yrs) | Body image generally poor; some (but not all) aspects of poor body image improved with intervention. |
| Becker-Hebly, et al. (2020). Psychosocial health in adolescents and young adults with gender dysphoria before and after gender-affirming medical interventions: a descriptive study from the Hamburg Gender Identity Service. European Child and Adolescent Psychiatry | 2020 | GERMANY<br>Gender Identity Service, University Medical Center Hamburg-Eppendorf | 0 | Included in Papers 2 (MH comorbidity) and 3 (treatment outcomes) |
| Biggs, M. (2020). Gender Dysphoria and Psychological Functioning in Adolescents Treated with GnRHa: Comparing Dutch and English Prospective Studies. Archives of Sexual Behavior, 49(7), 2231–2236. | 2020 | Secondary data: NETHERLANDS and UK | 5<br>No original data | Useful critique of existing data: compares Dutch and English samples |
| Bonifacio, J. H., Maser, C., Stadelman, K., & Palmert, M. (2019). Management of gender dysphoria in adolescents in primary care. Cmaj, 191(3), E69-E75. | 2019 | N/A–review paper<br>(Authors based in Toronto, CANADA) | 5 | Increase in cases will mean primary care services have to be prepared |
| Bradley, S. J. (1978). Gender identity problems of children and adolescents: The establishment of a special clinic. The Canadian Psychiatric Association Journal / La Revue de l'Association des psychiatres du Canada, 23(3), 175–183. | 1978 | CANADA<br>Toronto, southwestern Ontario | 3<br>Small N. | No objective data–clinical impressions only.<br>Interesting early paper on GD and Toronto clinical service. |

*(Continued)*

**Table 5.** (Continued)

| Reference | Date | Location & setting | Reason for exclusion | Notes |
|---|---|---|---|---|
| Brocksmith, V. M., Alradadi, R. S., Chen, M., & Eugster, E. A. (2018). Baseline characteristics of gender dysphoric youth. Journal of Pediatric Endocrinology and Metabolism, 31(12), 1367–1369. | 2018 | USA<br>pediatric endocrine clinic, Riley Hospital for Children, Indianapolis | 1<br>No separate data on adolescents (includes ≥18 yrs) | High proportion of NF to NM<br>50% overweight / obese<br>Anxiety higher in NF<br>Some indication of adolescent-onset being more common in recent (post-2014) participants |
| Bui, H. N., Schagen, S. E. E., Klink, D. T., Delemarre-Van De Waal, H. A., Blankenstein, M. A., & Heijboer, A. C. (2013). Salivary testosterone in female-To-male transgender adolescents during treatment with intra-muscular injectable testosterone esters. Steroids, 78(1), 91–95. | 2013 | NETHERLANDS<br>VU Medical Center, Amsterdam | 1, 3 | Technical paper focused on novel method of measuring salivary testosterone levels. |
| Burke, S. M., Kreukels, B. P. C., Cohen-Kettenis, P. T., Veltman, D. J., Klink, D. T., & Bakker, J. (2016). Male-typical visuospatial functioning in gynephilic girls with gender dysphoria—Organizational and activational effects of testosterone. Journal of Psychiatry and Neuroscience, 41(6), 395–404. | 2016 | NETHERLANDS<br>Center of Expertise on Gender Dysphoria (CEGD), Amsterdam | 3 | Small selected sample, unlikely to be representative.<br>Psychiatric disorder an exclusion criterion. |
| Butler, G., De Graaf, N., Wren, B., & Carmichael, P. (2018). Assessment and support of children and adolescents with gender dysphoria. Archives of Disease in Childhood, 103(7), 631–636. | 2018 | UK<br>Gender Identity Development Service, Tavistock, London | 5<br>Commissioned review | Some data on age and gender |
| Calzo, J. P., & Blashill, A. J. (2018). Child Sexual Orientation and Gender Identity in the Adolescent Brain Cognitive Development Cohort Study. JAMA Pediatrics, 172(11), 1090–1092. | 2018 | USA<br>San Diego (Adolescent Brain Cognitive Development (ABCD) study) | 2, 4 | Self-identification of sexuality and gender status, with very small N identifying as transgender. Sample aged 9–10 yrs. |
| Chen, D., Simons, L., Johnson, E. K., Lockart, B. A., & Finlayson, C. (2017). Fertility Preservation for Transgender Adolescents. Journal of Adolescent Health, 61(1), 120–123. | 2017 | USA<br>Gender & Sex Development Program (GSDP), Ann & Robert H. Lurie Children's Hospital of Chicago | 3 | Very small sample. 11/13 were adolescents. Sample was 11 adolescents presenting specifically for fertility preservation. |
| Chodzen, G., Hidalgo, M. A., Chen, D., & Garofalo, R. (2019). Minority Stress Factors Associated With Depression and Anxiety Among Transgender and Gender-Nonconforming Youth. Journal of Adolescent Health, 64(4), 467–471. | 2019 | USA<br>Division of Adolescent Medicine, Ann & Robert H. Lurie Children's Hospital of Chicago* | 2<br>Sample self-identified as transgender and gender-nonconforming (TGNC) | High levels of anxiety and depression. |
| Clark, T. C., Lucassen, M. F. G., Bullen, P., Denny, S. J., Fleming, T. M., Robinson, E. M., & Rossen, F. V. (2014). The health and well-being of transgender high school students: Results from the New Zealand adolescent health survey (youth'12). Journal of Adolescent Health, 55(1), 93–99. | 2014 | NEW ZEALAND<br>National population-based survey | 2<br>Survey data. Participants self-identify as transgender. | Adolescents identifying as transgender have considerable health and wellbeing needs relative to their peers. |
| Cohen, L., De Ruiter, C., Ringelberg, H., & Cohen-Kettenis, P. T. (1997). Psychological functioning of adolescent transsexuals: Personality and psychopathology. Journal of Clinical Psychology, 53(2), 187–196. | 1997 | NETHERLANDS<br>VU Medical Center, Amsterdam | 1 | Mean age 17.2±1.81<br>Rorschach methodology–not useful for our review |

*(Continued)*

**Table 5.** (Continued)

| Reference | Date | Location & setting | Reason for exclusion | Notes |
|---|---|---|---|---|
| Cohen-Kettenis, P. T., & Van Goozen, S. H. M. (1997). Sex reassignment of adolescent transsexuals: A follow-up study. Journal of the American Academy of Child and Adolescent Psychiatry, 36 (2), 263–271. | 1997 | NETHERLANDS University Medical Centre, Utrecht (moved to VUMC / CEGD in 2002). | 1, 3 | GD resolved at post-surgery follow-up. No expression of regret. |
| Coolidge, F. L., Thede, L. L., & Young, S. E. (2002). The heritability of gender identity disorder in a child and adolescent twin sample. Behavior Genetics, 32(4), 251–257. | 2002 | USA Colorado Springs, Colorado | 2 | Twin methdology to investigate heritability. |
| Day, J. K., Fish, J. N., Perez-Brumer, A., Hatzenbuehler, M. L., & Russell, S. T. (2017). Transgender Youth Substance Use Disparities: Results From a Population-Based Sample. Journal of Adolescent Health, 61(6), 729–735. doi:10.1016/j.jadohealth.2017.06.024 | 2017 | USA 2013–2015 Biennial Statewide California Student Survey | 2 Survey data. Participants self-identify as transgender. | Transgender youth at increased risk for substance misuse. Some psychosocial factors may mediate this. |
| de Graaf, et al. (2018). Sex ratio in children and adolescents referred to the Gender Identity Development Service in the UK (2009–2016). Archives of Sexual Behavior, 47(5), 1301–1304. doi:10.1007/s10508-018-1204-9 | 2018 | UK Gender Identity Development Service, Tavistock, London | 2 Referrals only–no GD clinical verification | Significant increases year-on-year, from only 39 adolescent referrals in 2009 to almost 1500 in 2016; average increase rate of referrals higher in NF. |
| de Graaf, N. M., Carmichael, P., Steensma, T. D., & Zucker, K. J. (2018). Evidence for a Change in the Sex Ratio of Children Referred for Gender Dysphoria: Data From the Gender Identity Development Service in London (2000–2017). Journal of Sexual Medicine, 15 (10), 1381–1383. | 2018 | UK Gender Identity Development Service, Tavistock, London | 4 | NM referred at younger age than NF. Recent increases have higher proportion NF (as observed in adolescent samples). |
| de Graaf, N. M., Manjra, II, Hames, A., & Zitz, C. (2019). Thinking about ethnicity and gender diversity in children and young people. Clinical Child Psychology & Psychiatry, 24(2), 291–303. | 2019 | UK Gender Identity Development Service, Tavistock, London | 2 Referrals only–no GD clinical verification | Black and minority ethnic groups were underrepresented. |
| De Pedro, K. T., Gilreath, T. D., Jackson, C., & Esqueda, M. C. (2017). Substance Use Among Transgender Students in California Public Middle and High Schools. The Journal of school health, 87 (5), 303–309. | 2017 | US 2013–2015 California Healthy Kids Survey (CHKS) | 2 Survey. Self-identified as transgender. | Transgender youth at increased risk for substance misuse. |
| De Vries, A. L. C., Noens, I. L. J., Cohen-Kettenis, P. T., Van Berckelaer-Onnes, I. A., & Doreleijers, T. A. (2010). Autism spectrum disorders in gender dysphoric children and adolescents. Journal of Autism and Developmental Disorders, 40 (8), 930–936. | 2010 | NETHERLANDS VU Medical Center, Amsterdam | 2 Not all participants diagnosed | Indication of association between GID and ASD |
| Delahunt, J. W., Denison, H. J., Sim, D. A., Bullock, J. J., & Krebs, J. D. (2018). Increasing rates of people identifying as transgender presenting to Endocrine Services in the Wellington region. New Zealand Medical Journal, 131(1468), 33–42. | 2018 | NEW ZEALAND Wellington Endocrine Service for Capital & Coast District Health Board | 1 No separate data on adolescents (includes ≥18 yrs) | Observed increase in referrals for people under 30, also increase in requests for female to male transition. |

*(Continued)*

**Table 5.** (Continued)

| Reference | Date | Location & setting | Reason for exclusion | Notes |
|---|---|---|---|---|
| Drummond, K. D., Bradley, S. J., Peterson-Badali, M., VanderLaan, D. P., & Zucker, K. J. (2018). Behavior Problems and Psychiatric Diagnoses in Girls with Gender Identity Disorder: A Follow-Up Study. Journal of Sex & Marital Therapy, 44(2), 172–187. | 2018 | CANADA Gender Identity Service, Center for Addiction and Mental Health, Toronto | 1, 6 Included some DSD; No separate data on adolescents | 12% (n = 3) of those referred in childhood showed persistent GID in adolescence / adulthood. Some indication of psychiatric vulnerability at follow-up, but a lot of variability. |
| Drummond, K. D., Bradley, S. J., Peterson-Badali, M., & Zucker, K. J. (2008). A follow-up study of girls with gender identity disorder. Developmental Psychology, 44(1), 34–45. | 2008 | CANADA Gender Identity Service, Center for Addiction and Mental Health, Toronto | 1 Follow-up took place from age 17; small n under 18 (and no distinct data reported) | As above |
| Durwood, L., McLaughlin, K. A., & Olson, K. R. (2017). Mental health and self-worth in socially transitioned transgender youth. Journal of the American Academy of Child & Adolescent Psychiatry, 56(2), 116–123. doi:10.1016/j.jaac.2016.10.016 | 2017 | NORTH AMERICA Multisite survey–TransYouth Project | 2 Survey data. Participants self-identify as transgender. | Socially transitioned young people had MH no worse than others their age |
| Getahun, D., Nash, R., Flanders, W. D., Baird, T. C., Becerra-Culqui, T. A., Cromwell, L., . . . Goodman, M. (2018). Cross-sex Hormones and Acute Cardiovascular Events in Transgender Persons: A Cohort Study. Annals of Internal Medicine, 169(4), 205–213. doi:10.7326/M17-2785 | 2018 | USA Kaiser Permanente sites in Georgia, northern California, southern California | 1 Adults only | Indication of increased risk of cardiovascular events in transfeminine participants. |
| Handler, T., Hojilla, J. C., Varghese, R., Wellenstein, W., Satre, D. D., & Zaritsky, E. (2019). Trends in Referrals to a Pediatric Transgender Clinic. Pediatrics, 144(5), 11. | 2019 | USA Kaiser Permanente Northern California health system | 2 No clinical verification of GD / GID. | Observed increase in referrals in recent years. Large proportion identify as transmasculine. Treatment needs varied by age group. |
| Hannema, S. E., Schagen, S. E. E., Cohen-Kettenis, P. T., & Delemarre-Van De Waal, H. A. (2017). Efficacy and safety of pubertal induction using 17beta-estradiol in transgirls. Journal of Clinical Endocrinology and Metabolism, 102(7), 2356–2363. | 2017 | NETHERLANDS Centre of Expertise on Gender Dysphoria (CEGD), Amsterdam | 1 No separate data on adolescents (includes ≥18 yrs) | Estradiol effective for pubertal induction in NM. |
| Hughes, S. K., VanderLaan, D. P., Blanchard, R., Wood, H., Wasserman, L., & Zucker, K. J. (2017). The Prevalence of Only-Child Status Among Children and Adolescents Referred to a Gender Identity Service Versus a Clinical Comparison Group. Journal of Sex & Marital Therapy, 43(6), 586–593. | 2017 | CANADA Gender Identity Service, CYFS, Toronto | 2, 4 Mean age of all groups <12 yrs. No apparent clinical verification of GD / GID. | Prevalence of only-child status not elevated in gender-referred children (compared to other clinical populations). |
| Janssen, A., Huang, H., & Duncan, C. (2016). Gender Variance among Youth with Autism Spectrum Disorders: A Retrospective Chart Review. Transgender Health, 1(1), 63–68. | 2016 | USA New York University Child Study Center | 2 Derived from clinical ASD sample. No clinical verification of GD / GID. | Participants with ASD diagnosis more likely to report gender variance on CBCL then CBCL normative sample. |

(*Continued*)

**Table 5.** (Continued)

| Reference | Date | Location & setting | Reason for exclusion | Notes |
|---|---|---|---|---|
| Jarin, J., Pine-Twaddell, E., Trotman, G., Stevens, J., Conard, L. A., Tefera, E., & Gomez-Lobo, V. (2017). Cross-sex hormones and metabolic parameters in adolescents with gender dysphoria. Pediatrics, 139 (5) (no pagination) (e20163173). | 2017 | USA Multi-site. MedStar Washington Hospital Center and Children's National Medical Center (both, Washington, DC). University of Maryland Medical Center, Baltimore. Cincinnati Children's Hospital Medical Center, Ohio. | 1 No separate data on adolescents (includes ≥18 yrs) | Testosterone use associated with increased hemoglobin and hematocrit, increased BMI, and lowered high-density lipoprotein levels. No significant change in those taking estrogen. |
| Jensen, R. K., Jensen, J. K., Simons, L. K., Chen, D., Rosoklija, I., & Finlayson, C. A. (2019). Effect of Concurrent Gonadotropin-Releasing Hormone Agonist Treatment on Dose and Side Effects of Gender-Affirming Hormone Therapy in Adolescent Transgender Patients. Transgender Health, 4(1), 300–303. | 2019 | USA Ann & Robert H. Lurie Children's Hospital of Chicago, Chicago, Illinois* | 0 | Included in Paper 3 (treatment outcomes) |
| Kaltiala, R., Bergman, H., Carmichael, P., de Graaf, N. M., Egebjerg Rischel, K., Frisen, L., . . . Waehre, A. (2020). Time trends in referrals to child and adolescent gender identity services: a study in four Nordic countries and in the UK. Nordic Journal of Psychiatry, 74(1), 40–44. | 2020 | DENMARK, FINLAND, NORWAY, SWEDEN, & the UK | 2 No clinical verification of GD / GID. | Comprehensive overview of referrals in 5 countries. Same pattern of increase, especially in NF, as in included papers. |
| Kaltiala, R., Heino, E., Tyolajarvi, M., & Suomalainen, L. (2020). Adolescent development and psychosocial functioning after starting cross-sex hormones for gender dysphoria. Nordic Journal of Psychiatry, 74(3), 213–219. | 2020 | FINLAND Tampere University Hospital, Department of Adolescent Psychiatry | 1 Mean age at assessment 18.1 | MH problems persisted during treatment–concluded that GD treatment not enough to address MH problems. |
| Kaltiala-Heino, R., Tyolajarvi, M., & Lindberg, N. (2019). Gender dysphoria in adolescent population: A 5-year replication study. Clinical Child Psychology and Psychiatry, 24(2), 379–387. | 2019 | FINLAND School survey in Tampere | 2 Survey data. No clinical verification of GD / GID. | Apparent increase in likely clinically-significant GD in adolescent population (2013–2017). |
| Katz-Wise, S. L., Ehrensaft, D., Vetters, R., Forcier, M., & Austin, S. B. (2018). Family Functioning and Mental Health of Transgender and Gender-Nonconforming Youth in the Trans Teen and Family Narratives Project. Journal of sex research, 55(4–5), 582–590. | 2018 | USA New England region–survey from range of services / organisations | 2 Survey data. No clinical verification of GD / GID. | MH concerns reported. Better family functioning (from young person's persepective) associated with better MH outcomes. |
| Khatchadourian, K., Amed, S., & Metzger, D. L. (2014). Clinical management of youth with gender dysphoria in Vancouver. Journal of Pediatrics, 164(4), 906–911. doi:10.1016/j.jpeds.2013.10.068 | 2014 | CANADA British Columbia Children's Hospital Transgender Program, Vancouver | 1 No separate data on adolescents (includes ≥18 yrs) | Median age at initiation of testosterone NF 17.3 years (range 13.7–19.8 years); median age at initiation of estrogen in NM 17.9 years (range 13.3–22.3 years). Intervnention appropriate in selected individuals with relevant clinical support. |
| Klein, D. A., Roberts, T. A., Adirim, T. A., Landis, C. A., Susi, A., Schvey, N. A., & Hisle-Gorman, E. (2019). Transgender Children and Adolescents Receiving Care in the US Military Health Care System. JAMA Pediatrics. | 2019 | USA Military Health System Data Repository | 1 No separate data on adolescents (includes ≥18 yrs) | Increase in service use 2010–2017. Prescriptions increased with higher parental rank. |

(*Continued*)

**Table 5.** (Continued)

| Reference | Date | Location & setting | Reason for exclusion | Notes |
|---|---|---|---|---|
| Kolbuck, V. D., Muldoon, A. L., Rychlik, K., Hidalgo, M. A., & Chen, D. (2019). Psychological functioning, parenting stress, and parental support among clinic-referred prepubertal gender-expansive children. Clinical Practice in Pediatric Psychology, 7(3), 254–266. doi:10.1037/cpp0000293 | 2019 | USA<br>Division of Adolescent Medicine, Ann & Robert H. Lurie Children's Hospital of Chicago* | 4<br>Sample <12 yrs. | Association between GD symptoms in ADHD (hyperactive-impulsive) and CD where parenting stress high. |
| Lawlis, S. M., Donkin, H. R., Bates, J. R., Britto, M. T., & Conard, L. A. E. (2017). Health Concerns of Transgender and Gender Nonconforming Youth and Their Parents Upon Presentation to a Transgender Clinic. Journal of Adolescent Health, 61(5), 642–648. doi:10.1016/j.jadohealth.2017.05.025 | 2017 | USA<br>'a transgender clinic at a large tertiary pediatric hospital in the Midwest.' Oklahoma University Children's Hospital* | 2 | 66.1% attending for first appointment NF |
| Lee, J. Y., Finlayson, C., Olson-Kennedy, J., Garofalo, R., Chan, Y.-M., Glidden, D. V., & Rosenthal, S. M. (2020). Low Bone Mineral Density in Early Pubertal Transgender/Gender Diverse Youth: Findings From the Trans Youth Care Study. Journal of the Endocrine Society, 4(9), 1–12. doi:10.1210/jendso/bvaa065 | 2020 | USA<br>Children's Hospital Los Angeles, Lurie Children's Hospital, Boston Children's Hospital, University of California San Francisco Benioff Children's Hospital | 0 | Included in Paper 3 (treatment outcomes) |
| Lobato, M. I., Koff, W. J., Schestatsky, S. S., Chaves, C. P. V., Petry, A., Crestana, T., . . . Henriques, A. A. (2007). Clinical characteristics, psychiatric comorbidities and sociodemographic profile of transsexual patients from an outpatient clinic in Brazil. International Journal of Transgenderism, 10(2), 69–77. doi:10.1080/15532730802175148 | 2007 | BRAZIL<br>Hospital de Clínicas de Porto Alegre | 1 | 42.7% had at least one psychiatric comorbidity |
| Lothstein, L. M. (1980). The adolescent gender dysphoric patient: An approach to treatment and management. Journal of Pediatric Psychology, 5(1), 93–109. | 1980 | USA<br>Case Western<br>Reserve University (CWRU) Gender Identity Clinic, Cleveland, Ohio | 1, 3 | Case series from over 40 years ago |
| Lynch, M. M., Khandheria, M. M., & Meyer, W. J., III. (2015). Retrospective study of the management of childhood and adolescent gender identity disorder using medroxyprogesterone acetate. International Journal of Transgenderism, 16(4), 201–208. doi:10.1080/15532739.2015.1080649 | 2015 | USA<br>Gender<br>Identity Clinic, University of Texas Medical Branch | 3<br>Case series | Medroxyprogesterone Acetate found to be effective and low-cost oral alternative to injectable or implant GnRH analogues. Response to treatment and compliance were favourable. |
| Manners, P. J. (2009). Gender identity disorder in adolescence: A review of the literature. Child and Adolescent Mental Health, 14(2), 62–68. | 2009 | UK (review)<br>Salomons Clinical Psychology Training Program, Canterbury Christ Church University | 5<br>Review | Review now out of date |
| May, T., Pang, K., & Williams, K. J. (2017). Gender variance in children and adolescents with autism spectrum disorder from the National Database for Autism Research. International Journal of Transgenderism, 18(1), 7–15. doi:10.1080/15532739.2016.1241976 | 2017 | USA<br>National Database for Autism Research | 2<br>Derived from clinical ASD sample.<br>No clinical verification of GD / GID. | Higher prevalence of gender variance in ASD sample compared to non-referred samples (but similar to other clinical samples). |

(*Continued*)

**Table 5.** (Continued)

| Reference | Date | Location & setting | Reason for exclusion | Notes |
|---|---|---|---|---|
| Millington, K., Liu, E., & Chan, Y. M. (2019). The utility of potassium monitoring in gender-diverse adolescents taking spironolactone. Journal of the Endocrine Society, 3(5), 1031–1038. | 2019 | USA Gender Management Service Program, Boston Children's Hospital | 1 Sample likely to include those ≥18 yrs | Hyperkalemia in patients taking spironolactone for gender transition rare. Routine electrolyte monitoring may be unnecessary. |
| Millington, K., Schulmeister, C., Finlayson, C., Grabert, R., Olson-Kennedy, J., Garofalo, R., . . . Chan, Y. M. (2020). Physiological and Metabolic Characteristics of a Cohort of Transgender and Gender-Diverse Youth in the United States. Journal of Adolescent Health, 67(3), 376–383. | 2020 | USA Children's Hospital Los Angeles/ University of Southern California, Boston Children's Hospital/Harvard Medical School, the Ann & Robert H. Lurie Children's Hospital of Chicago/ Northwestern University, Benioff Children's Hospital/University of California San Francisco | 4 Sample 1 aged 8–14 (so mostly under 12s); sample 2 aged 12–20 (so included over 18s). | Description of baseline metabolic characteristics–will be useful to see cohorts followed up. |
| Munck, E. T. (2000). A retrospective study of adolescents visiting a Danish clinic for sexual disorders. International Journal of Adolescent Medicine and Health, 12(2–3), 215–222. doi:10.1515/ IJAMH.2000.12.2–3.215 | 2000 | DENMARK Sexological Clinic, Copenhagen University Hospital | 1 Mean age over 20 years | Description of cohort 1686–1995. Up to age 16, majority were NM. From age 17, majority were NF. |
| Nahata, L., Tishelman, A. C., Caltabellotta, N. M., & Quinn, G. P. (2017). Low Fertility Preservation Utilization Among Transgender Youth. Journal of Adolescent Health, 61(1), 40–44. | 2017 | USA Division of Endocrinology, Department of Pediatrics, Nationwide Children's Hospital, The Ohio State University College of Medicine, Columbus, Ohio* | 5 Same sample already described (Nahata et al., 2017) | See epidemiological data in main review (Nahata et al., 2017). |
| Neyman, A., Fuqua, J. S., & Eugster, E. A. (2019). Bicalutamide as an Androgen Blocker With Secondary Effect of Promoting Feminization in Male-to-Female Transgender Adolescents. Journal of Adolescent Health, 64(4), 544–546. | 2019 | USA Pediatric Endocrine Clinic, Riley Hospital for Children, Indiana | 1, 3 Where adolescent data separated, includes very small sample (case series) | Evidence that bicalutamide may be viable alternative to gonadotrophin-releasing hormone analogues in NM ready to transition |
| O'Bryan, J., Scribani, M., Leon, K., Tallman, N., Wolf-Gould, C., Wolf-Gould, C., & Gadomski, A. (2020). Health-related quality of life among transgender and gender expansive youth at a rural gender wellness clinic. Quality of Life Research, 29(6), 1597–1607. | 2020 | USA The Gender Wellness Center (GWC) of the Bassett Health-care Network, New York | 1 Upper age limit 25 yrs. No meaningful separate data on those under 18 yrs. | Poor MH reported (relative to general population). Long term follow-up needed. |
| Olson, J., Schrager, S. M., Belzer, M., Simons, L. K., & Clark, L. F. (2015). Baseline Physiologic and Psychosocial Characteristics of Transgender Youth Seeking Care for Gender Dysphoria. Journal of Adolescent Health, 57(4), 374–380. doi:10.1016/j.jadohealth.2015.04.027 | 2015 | USA Center for Transyouth Health and Development, Children's Hospital Los Angeles, California | 1 | Awareness of gender incongruity from young age (mean 8.3 yrs). Physiological characteristics within normal ranges. 35% experiencing depression; 51% contemplated suicide; 30% attempted suicide. |
| Olson-Kennedy, J., Okonta, V., Clark, L. F., & Belzer, M. (2018). Physiologic Response to Gender-Affirming Hormones Among Transgender Youth. Journal of Adolescent Health, 62(4), 397–401. | 2018 | USA Center for Transyouth Health and Development, Children's Hospital Los Angeles, California | 1 | Use of gender affirming hormones not associated with clinically significant changes in metabolic parameters. May not need to frequently monitor transgender adolescents. |

(*Continued*)

**Table 5.** (Continued)

| Reference | Date | Location & setting | Reason for exclusion | Notes |
|---|---|---|---|---|
| Olson-Kennedy, J., Warus, J., Okonta, V., Belzer, M., & Clark, L. F. (2018). Chest Reconstruction and Chest Dysphoria in Transmasculine Minors and Young Adults: Comparisons of Nonsurgical and Postsurgical Cohorts. JAMA Pediatrics, 172(5), 431–436. doi:10.1001/jamapediatrics.2017.5440 | 2018 | USA Center for Transyouth Health and Development, Children's Hospital Los Angeles, California | 1 | Mean age at chest surgery 17.5 (2.4) years. 49% younger than 18 years. All postsurgical participants (n = 68) felt surgery had been a good decision. Loss of nipple sensation most common side-effect. |
| Ospina, N. M. S., Maraka, S., Rodriguez-Gutierrez, R., Davidge-Pitts, C. J., Nippoldt, T. B., & Murad, M. H. (2016). Effect of sex steroids on the bone health of transgender individuals: A systematic review and meta-analysis. Endocrine Reviews. Conference: 98th Annual Meeting and Expo of the Endocrine Society, ENDO, 37(2 Supplement 1). | 2016 | International (researcher based in USA) | 5 Systematic review | Bone mineral density (lumbar spine) increased in NM 12–24 months after initiating feminising hormone therapy. No changes in NF with masculinising therapy. |
| Pakpoor, J., Wotton, C. J., Schmierer, K., Giovannoni, G., & Goldacre, M. J. (2016). Gender identity disorders and multiple sclerosis risk: A national record-linkage study. Multiple Sclerosis, 22(13), 1759–1762. | 2016 | UK English national Hospital Episode Statistics (HES) and mortality data | 5 No useable data for our questions | |
| Pang, K. C., de Graaf, N. M., Chew, D., Hoq, M., Keith, D. R., Carmichael, P., & Steensma, T. D. (2020). Association of Media Coverage of Transgender and Gender Diverse Issues With Rates of Referral of Transgender Children and Adolescents to Specialist Gender Clinics in the UK and Australia. JAMA Network Open, 3(7), e2011161. | 2020 | UK & Australia | 2 Data from referral only: GD not clinically verified. | Evidence of association between media coverage and number of new referrals to services. |
| Perez-Brumer, A., Day, J. K., Russell, S. T., & Hatzenbuehler, M. L. (2017). Prevalence and Correlates of Suicidal Ideation Among Transgender Youth in California: Findings From a Representative, Population-Based Sample of High School Students. Journal of the American Academy of Child & Adolescent Psychiatry, 56(9), 739–746. doi:10.1016/j.jaac.2017.06.010 | 2017 | USA California Healthy Kids Survey | 2 | Transgender youth had 2.99 higher odds of reporting past-year suicidal ideation than non-transgender youth. |
| Perl, L., Segev-Becker, A., Israeli, G., Elkon-Tamir, E., & Oren, A. (2020). Blood Pressure Dynamics After Pubertal Suppression with Gonadotropin-Releasing Hormone Analogs Followed by Testosterone Treatment in Transgender Male Adolescents: A Pilot Study. LGBT health, 7(6), 340–344. | 2020 | ISRAEL Israeli Pediatric Gender Dysphoria Clinic, Dana-Dwek Children's Hospital, Tel Aviv Sourasky Medical Center | 0 | Included in Paper 3 (treatment outcomes) |
| Peterson, C. M., Matthews, A., Copps-Smith, E., & Conard, L. A. (2017). Suicidality, Self-Harm, and Body Dissatisfaction in Transgender Adolescents and Emerging Adults with Gender Dysphoria. Suicide & Life-Threatening Behavior, 47(4), 475–482. | 2017 | USA Cincinnati Children's Hospital Medical Center Transgender Clinic, Ohio | 1 | 30.3% transgender youth reported history of ≥1 suicide attempt; 41.8% history self-injury. Higher suicidality in NF than NM. |

(*Continued*)

**Table 5.** (Continued)

| Reference | Date | Location & setting | Reason for exclusion | Notes |
|---|---|---|---|---|
| Quinn, V. P., Nash, R., Hunkeler, E., Contreras, R., Cromwell, L., Becerra-Culqui, T. A., . . . Goodman, M. (2017). Cohort profile: Study of Transition, Outcomes and Gender (STRONG) to assess health status of transgender people. BMJ Open, 7(12), e018121. | 2017 | USA Kaiser-Permanente records, California and Georgia | 1 No useable data by age group | Useable data (proportion NM to NF) described in Becerra-Culqui et al. (2018) |
| Reisner, S. L., Biello, K. B., Hughto, J. M. W., Kuhns, L., Mayer, K. H., Garofalo, R., & Mimiaga, M. J. (2016). Psychiatric diagnoses and comorbidities in a diverse, multicity cohort of young transgender women: Baseline Findings from Project LifeSkills. JAMA Pediatrics, 170(5), 481–486. | 2016 | USA: Chicago and Boston–Project LifeSkills | 1, 2 No separate data on adolescents (includes ≥18 yrs). Current GD not an inclusion criterion. | 41.5% of sample of NM had 1 or more mental health or substance dependence diagnoses; 20% had 2 or more comorbid psychiatric diagnoses |
| Reisner, S. L., Vetters, R., Leclerc, M., Zaslow, S., Wolfrum, S., Shumer, D., & Mimiaga, M. J. (2015). Mental health of transgender youth in care at an adolescent Urban community health center: A matched retrospective cohort study. Journal of Adolescent Health, 56 (3), 274–279. | 2015 | US Sidney Borum Jr Health Center, Boston | 1 No separate data on adolescents (includes ≥18 yrs; mean age 19.6 ±3.0). | Increased risk of MH problems in transgender vs cisgender youth. No difference by natal gender. |
| Rider, G. N., Berg, D., Pardo, S. T., Olson-Kennedy, J., Sharp, C., Tran, K. M., . . . Keo-Meier, C. L. (2019). Using the Child Behavior Checklist (CBCL) with transgender/gender nonconforming children and adolescents. Clinical Practice in Pediatric Psychology, 7(3), 291–301. doi:10.1037/cpp0000296 | 2019 | USA Trans Youth and Family Allies project (national) | 2 Survey | No significant impact in use of gendered scoring templates on CBCL |
| Roberts, A. L., Rosario, M., Slopen, N., Calzo, J. P., & Austin, S. B. (2013). Childhood gender nonconformity, bullying victimization, and depressive symptoms across adolescence and early adulthood: An 11-year longitudinal study. Journal of the American Academy of Child & Adolescent Psychiatry, 52(2), 143–152. doi:10.1016/j.jaac.2012.11.006 | 2013 | USA Growing Up Today Study (GUTS) (national) | 1, 2 Survey Age range 12–30 | Large longitudinal cohort. Association between gender nonconformity and depressive symptoms. |
| Röder, M., Barkmann, C., Richter-Appelt, H., Schulte-Markwort, M., Ravens-Sieberer, U., & Becker, I. (2018). Health-related quality of life in transgender adolescents: Associations with body image and emotional and behavioral problems. International Journal of Transgenderism, 19(1), 78–91. doi:10.1080/15532739.2018.1425649 | 2018 | GERMANY Hamburg Gender Identity Service for Children and Adolescents | 1 Upper age 18.2 | Health related quality of life (HRQoL) generally poorer in ttransgender adolescents vs normative scores. Body satisfaction and internalising problems significant predictors of HRQoL. |
| Schagen, S. E. E., Cohen-Kettenis, P. T., Delemarre-van de Waal, H. A., & Hannema, S. E. (2016). Efficacy and Safety of Gonadotropin-Releasing Hormone Agonist Treatment to Suppress Puberty in Gender Dysphoric Adolescents. Journal of Sexual Medicine, 13(7), 1125–1132. | 2016 | NETHERLANDS Centre of Expertise on Gender Dysphoria (CEGD), Amsterdam | 1 Upper age 18.6 | Triptorelin effective in suppressing puberty. Routine monitoring of gonadotropins, sex steroids, creatinine, and liver function may not be necessary. |

(*Continued*)

**Table 5.** (Continued)

| Reference | Date | Location & setting | Reason for exclusion | Notes |
|---|---|---|---|---|
| Schagen, et al. (2018). Changes in Adrenal Androgens During Puberty Suppression and Gender-Affirming Hormone Treatment in Adolescents With Gender Dysphoria. Journal of Sexual Medicine, 15(9), 1357–1363. | 2018 | NETHERLANDS Centre of Expertise on Gender Dysphoria (CEGD), Amsterdam | 1 Max age 18.6 | No harmful effects of treatment of GnRHa and gender affirming hormone treatment on adrenal androgen levels were found during approximately 4 years of follow-up. |
| Schagen, S. E. E., Wouters, F. M., Cohen-Kettenis, P. T., Gooren, L. J., & Hannema, S. E. (2020). Bone Development in Transgender Adolescents Treated With GnRH Analogues and Subsequent Gender-Affirming Hormones. Journal of Clinical Endocrinology & Metabolism, 105(12), 01. | 2020 | NETHERLANDS Centre of Expertise on Gender Dysphoria (CEGD), Amsterdam | 0 | Included in Paper 3 (treatment outcomes) |
| Shields, J. P., Cohen, R., Glassman, J. R., Whitaker, K., Franks, H., & Bertolini, I. (2013). Estimating population size and demographic characteristics of lesbian, gay, bisexual, and transgender youth in middle school. Journal of Adolescent Health, 52(2), 248–250. | 2013 | USA Youth Risk Behavior Survey (YRBS), San Francisco, California | 2 Survey | 1.3% of middle school students identified as transgender |
| Shumer, D. E., Reisner, S. L., Edwards-Leeper, L., & Tishelman, A. (2016). Evaluation of Asperger Syndrome in Youth Presenting to a Gender Dysphoria Clinic. LGBT Health, 3(5), 387–390. | 2016 | USA Boston Children's Hospital | 1, 3 Sample included adults (max age 20 yrs). Small sub-sample: only 6 aged 12–18 yrs ASQ>80 | 9/39 (23.1%) GD participants had indication of Asperger Syndrome. |
| Skagerberg, E., Di Ceglie, D., & Carmichael, P. (2015). Brief Report: Autistic Features in Children and Adolescents with Gender Dysphoria. Journal of Autism and Developmental Disorders, 45(8), 2628–2632. | 2015 | UK Gender Identity Development Service, Tavistock, London | 2 No GD dx reported | Positive association between SRS scores and ASD symptoms. |
| Skagerberg, E., Parkinson, R., & Carmichael, P. (2013). Self-harming thoughts and behaviors in a group of children and adolescents with gender dysphoria. International Journal of Transgenderism, 14(2), 86–92. doi:10.1080/15532739.2013.817321 | 2013 | UK Gender Identity Development Service, Tavistock, London | 2 No GD dx reported | 24% self-harmed, 14% had thoughts of self-harming, suicide attempts indicated in 10% prior to attending GIDS. Thoughts of self-harm more common in NM, actual self-harm more common in NF. |
| Smith, Y. L. S., Van Goozen, S. H. M., & Cohen-Kettenis, P. T. (2001). Adolescents with gender identity disorder who were accepted or rejected for sex reassignment surgery: A prospective follow-up study. Journal of the American Academy of Child and Adolescent Psychiatry, 40(4), 472–481. | 2001 | NETHERLANDS University Medical Centre, Utrecht (moved to VUmc / CEGD in 2002). | 1 No separate data on adolescents (includes ≥18 yrs) | Group no longer GD after sex reassignment surgery. No one expressed regrets. Non-treated group showed some improvement in MH, but also had 'more dysfunctional psychological profile'. |
| Smith, Y. L. S., Van Goozen, S. H. M., Kuiper, A. J., & Cohen-Kettenis, P. T. (2005). Sex reassignment: Outcomes and predictors of treatment for adolescent and adult transsexuals. Psychological Medicine, 35(1), 89–99. | 2005 | NETHERLANDS VU University Medical Centre, Amsterdam (VUmc) or University Medical Centre, Utrecht (UMCU) | 1 No separate data on adolescents (includes ≥18 yrs) | Group no longer GD after sex reassignment surgery. |

(*Continued*)

**Table 5.** (Continued)

| Reference | Date | Location & setting | Reason for exclusion | Notes |
|---|---|---|---|---|
| Spack, N. P., Edwards-Leeper, L., Feldman, H. A., Leibowitz, S., Mandel, F., Diamond, D. A., & Vance, S. R. (2012). Children and adolescents with gender identity disorder referred to a pediatric medical center. Pediatrics, 129(3), 418–425. | 2012 | USA GeMS clinic, Endocrine Division, Children's Hospital Boston | 1 No separate data on adolescents (includes ≥18 yrs) | 44.3% had significant psychiatric history. Noted four-fold increase in presentations of GID following establishment of specialist service. |
| Steensma, T. D., & Cohen-Kettenis, P. T. (2015). More than two developmental pathways in children with gender dysphoria? Journal of the American Academy of Child and Adolescent Psychiatry, 54(2), 147–148. | 2015 | NETHERLANDS Centre of Expertise on Gender Dysphoria (CEGD), Amsterdam | 1 Letter to the editor: cannot determine if cohort already described in included papers. | Posits distinction between 'persisters' and 'persisters after interruption'. |
| Steensma, T. D., McGuire, J. K., Kreukels, B. P. C., Beekman, A. J., & Cohen-Kettenis, P. T. (2013). Factors associated with desistence and persistence of childhood gender dysphoria: A quantitative follow-up study. Journal of the American Academy of Child and Adolescent Psychiatry, 52 (6), 582–590. | 2013 | NETHERLANDS Centre of Expertise on Gender Dysphoria (CEGD), Amsterdam | 1 No separate data on adolescents (includes ≥18 yrs) | Persistence of GD associated with early intensity of symptoms and being NF. Noted differing presentation by natal gender. |
| Steensma, T. D., Zucker, K. J., Kreukels, B. P. C., VanderLaan, D. P., Wood, H., Fuentes, A., & Cohen-Kettenis, P. T. (2014). Behavioral and emotional problems on the Teacher's Report Form: A cross-national, cross-clinic comparative analysis of gender dysphoric children and adolescents. Journal of Abnormal Child Psychology, 42(4), 635–647. doi:10.1007/s10802-013-9804-2 | 2014 | NETHERLANDS Centre of Expertise on Gender Dysphoria (CEGD), Amsterdam AND CANADA Gender Identity Service, CYFS, Toronto | 1 No separate data on adolescents (includes ≥18 yrs) (Adolescent subgroup likely to include those up to age 19–20, based on means / SDs given) | Teacher-reported emotional and behavioral problems greater in adolescents than in children. Internalising and externalising problems greater in NM than NF. Canadian sample had greater emotional and behavioural problems than Dutch sample. |
| Stoffers, I. E., de Vries, M. C., & Hannema, S. E. (2019). Physical changes, laboratory parameters, and bone mineral density during testosterone treatment in adolescents with gender dysphoria. Journal of Sexual Medicine, 16(9), 1459–1468. | 2019 | NETHERLANDS Department of Pediatrics, Leiden University Medical Centre, Leiden* | 0 | Included in Papers 3 (treatment outcomes) |
| Strang, J. F., Powers, M. D., Knauss, M., Sibarium, E., Leibowitz, S. F., Kenworthy, L., . . . Anthony, L. G. (2018). "They Thought It Was an Obsession": Trajectories and Perspectives of Autistic Transgender and Gender-Diverse Adolescents. Journal of Autism and Developmental Disorders, 48(12), 4039–4055. | 2018 | USA Center for Neuroscience and Behavioral Medicine, Children's National Health System, Washington, DC* | 1 No separate data on adolescents (includes ≥18 yrs) | Useful qualitative study on young people's perspectives. No relevant data for our research questions. |
| Sumia, M., Lindberg, N., Tyolajarvi, M., & Kaltiala-Heino, R. (2016). Early pubertal timing is common among adolescent girl-to-boy sex reassignment applicants. European Journal of Contraception and Reproductive Health Care, 21(6), 483–485. | 2016 | FINLAND Tampere University Hospital, Tampere Helsinki University Hospital, Helsinki | 2 | GD in adolescence associated with early pubertal timing in NF |

(*Continued*)

**Table 5.** (Continued)

| Reference | Date | Location & setting | Reason for exclusion | Notes |
|---|---|---|---|---|
| Sumia, M., Lindberg, N., Tyolajarvi, M., & Kaltiala-Heino, R. (2017). Current and recalled childhood gender identity in community youth in comparison to referred adolescents seeking sex reassignment. Journal of Adolescence, 56, 34–39. | 2017 | FINLAND<br>School survey in Tampere & clinically referred population: Tampere and Helsinki | 1<br>No separate data on adolescents (includes ≥18 yrs) (Adolescent subgroup likely to include those 18+, based on means / SDs given) | Interesting exploration of gender identity in GD and community samples. No data directly relevant to our research questions. |
| Tack, et al. (2016). Consecutive lynestrenol and cross-sex hormone treatment in biological female adolescents with gender dysphoria: A retrospective analysis. Hormone Research in Paediatrics, 86 (Supplement 1), 268–269. | 2016 | BELGIUM<br>Division of Pediatric Endocrinology, Ghent University* | 0 | Included in Papers 2 (MH comorbidity) and 3 (treatment outcomes) |
| Tack, L. J. W., Heyse, R., Craen, M., Dhondt, K., Bossche, H. V., Laridaen, J., & Cools, M. (2017). Consecutive Cyproterone Acetate and Estradiol Treatment in Late-Pubertal Transgender Female Adolescents. Journal of Sexual Medicine, 14(5), 747–757. | 2017 | BELGIUM<br>Division of Pediatric Endocrinology, Ghent University* | 0 | Included in Papers 2 (MH comorbidity) and 3 (treatment outcomes) |
| Tollit, M. A., Pace, C. C., Telfer, M., Hoq, M., Bryson, J., Fulkoski, N., . . . Pang, K. C. (2019). What are the health outcomes of trans and gender diverse young people in Australia? Study protocol for the Trans20 longitudinal cohort study. BMJ Open, 9(11), e032151. | 2019 | AUSTRALIA<br>Royal Children's Hospital Gender Service (RCHGS), Melbourne | 5<br>Cohort description | Protocol paper only. |
| Twist, J., & de Graaf, N. M. (2019). Gender diversity and non-binary presentations in young people attending the United Kingdom's National Gender Identity Development Service. Clinical Child Psychology and Psychiatry, 24(2), 277–290. | 2019 | UK<br>Gender Identity Development Service, Tavistock, London | 2<br>No clinical verification of GD / GID–new questionnaire completed at presentation. | Useful in relation to prevalence of different types of gender self-identification at clinics |
| van der Miesen, A. I. R., Hurley, H., Bal, A. M., & de Vries, A. L. C. (2018). Prevalence of the Wish to be of the Opposite Gender in Adolescents and Adults with Autism Spectrum Disorder. Archives of Sexual Behavior, 47(8), 2307–2317. | 2018 | NETHERLANDS<br>Centre of Expertise on Gender Dysphoria (CEGD), Amsterdam | 2<br>No clinical verification of GD / GID–endorsement of single item on YSR only. | Significantly more adolescents (6.5%) with ASD endorsed item expressing wish to be the opposite gender compared to the general population (3–5%). NF endorsed more then NM. Adolescents with ASD who endorsed gender item had higher YSR scores (poorer MH). No association with any specific subdomain of ASD. |
| van der Miesen, A. I. R., Steensma, T. D., de Vries, A. L. C., Bos, H., & Popma, A. (2020). Psychological Functioning in Transgender Adolescents Before and After Gender-Affirmative Care Compared With Cisgender General Population Peers. Journal of Adolescent Health, 66(6), 699–704. | 2020 | NETHERLANDS<br>Centre of Expertise on Gender Dysphoria (CEGD), Amsterdam | 2 (group 1)<br>1 (group 2) | Poor MH in referrals. MH in those receiving treatment was similar to general population sample. |
| Van Donge, N., Schvey, N. A., Roberts, T. A., & Klein, D. A. (2019). Transgender Dependent Adolescents in the U.S. Military Health Care System: Demographics, Treatments Sought, and Health Care Service Utilization. Military medicine, 184(5–6), e447-e454. | 2019 | USA<br>Transgender and gender-diverse clinic for children of military personnel | 1<br>No separate data on adolescents (includes ≥18 yrs) | Mean age at first gender-related visit 14.5 years (SD 3.2). History of self-harm (42%), suicidal ideation (70%), suicide attempt (21%), and psychiatric hospitalisation (33%). |

*(Continued)*

**Table 5.** (Continued)

| Reference | Date | Location & setting | Reason for exclusion | Notes |
|---|---|---|---|---|
| Vlot, M. C., Klink, D. T., den Heijer, M., Blankenstein, M. A., Rotteveel, J., & Heijboer, A. C. (2017). Effect of pubertal suppression and cross-sex hormone therapy on bone turnover markers and bone mineral apparent density (BMAD) in transgender adolescents. Bone, 95, 11–19. | 2017 | NETHERLANDS Centre of Expertise on Gender Dysphoria (CEGD), Amsterdam | 1 No separate data on adolescents (includes ≥18 yrs) | Suppressing puberty by GnRHa leads to a decrease of bone turnover markers (BTMs) in transgender adolescents, but added value of evaluating BTMs in transgender adolescents seems to be limited and requires further research. DXA-scans remain important in follow-up. |
| Wallien, M. S. C., & Cohen-Kettenis, P. T. (2008). Psychosexual outcome of gender-dysphoric children. Journal of the American Academy of Child and Adolescent Psychiatry, 47(12), 1413–1423. | 2008 | NETHERLANDS VU University Medical Center (forerunner to CEGD), Amsterdam | 4 Onset <12 years | Most children with GD were not GD after puberty. Those with persistent GD had more intense GD in childhood than those desisting. |
| Wallien, M. S. C., Swaab, H., & Cohen-Kettenis, P. T. (2007). Psychiatric comorbidity among children with gender identity disorder. Journal of the American Academy of Child and Adolescent Psychiatry, 46(10), 1307–1314. | 2007 | NETHERLANDS VU University Medical Center (forerunner to CEGD), Amsterdam | 4 all < 12 years | 52% of GID children had one or more other diagnoses. Internalising problems more common (37%) than externalising (23%). 31% of GID group had anxiety disorder. |
| Watson, R. J., Veale, J. F., & Saewyc, E. M. (2017). Disordered eating behaviors among transgender youth: Probability profiles from risk and protective factors. International Journal of Eating Disorders, 50(5), 515–522. doi:10.1002/eat.22627 | 2017 | CANADA Canadian Trans Youth Health Survey (national) | 2 Survey | High rates of eating disorder behaviour among self-identified transgender youth. Risk for eating disordered behaviours linked to enacted stigma and violence exposure, and offset by social supports. |
| Wood, H., Sasaki, S., Bradley, S. J., Singh, D., Fantus, S., Owen-Anderson, A., . . . Zucker, K. J. (2013). Patterns of referral to a gender identity service for children and adolescents (1976–2011): age, sex ratio, and sexual orientation. Journal of Sex & Marital Therapy, 39(1), 1–6. | 2013 | CANADA Gender Identity Service, CYFS, Toronto | 1, 4 No separate data on adolescents (includes <12 yrs and ≥18 yrs) | Sharp increase in adolescent referrals in 2004–2007 time period (compared to 1976–2003), continued into 2008–2011 time block. NF exceeded NM in most recent (2008–2011) cohort. |
| Yadegarfard, M., Ho, R., & Bahramabadian, F. (2013). Influences on loneliness, depression, sexual-risk behaviour and suicidal ideation among Thai transgender youth. Culture, Health & Sexuality, 15(6), 726–737. doi:10.1080/13691058.2013.784362 | 2013 | THAILAND Survey through range of organisations, via Rainbow Sky Association, Bangkok | 1, 2 | Education level (did not graduate high school) associated with less loneliness but more depression than those with some university credit. |
| Zou, Y., Szczesniak, R., Teeters, A., Conard, L. A. E., & Grossoehme, D. H. (2018). Documenting an epidemic of suffering: low health-related quality of life among transgender youth. Quality of Life Research, 27(8), 2107–2115. | 2018 | USA Transgender Clinic of the Division of Adolescent and Transition Medicine, Cincinnati Children's Hospital Medical Center, Ohio | 1 | Transgender / gender non-conforming youth reported low health related quality of life across all domains. Most were significantly lower than healthy peers or peers with chronic diseases. |
| Zucker, K. J., Owen, A., Bradley, S. J., & Ameeriar, L. (2002). Gender-dysphoric children and adolescents: A comparative analysis of demographic characteristics and behavioral problems. Clinical Child Psychology and Psychiatry, 7(3), 398–411. doi:10.1177/1359104502007003007 | 2002 | CANADA Gender Identity Service, CYFS, Toronto | 1, 4 No separate data on adolescents (includes <12 yrs and ≥18 yrs) | 84.7% of adolescents had CBCL sum score in clinical range (>90th centile). Scores strongly predicted by peer relations scale (i.e., poor peer relations predicted behavioural psychopathology). |

(Continued)

**Table 5.** (Continued)

| Reference | Date | Location & setting | Reason for exclusion | Notes |
|---|---|---|---|---|
| Zucker, K. J., Bradley, S. J., Owen-Anderson, A., Kibblewhite, S. J., & Cantor, J. M. (2008). Is gender identity disorder in adolescents coming out of the closet? Journal of Sex and Marital Therapy, 34(4), 287–290. | 2008 | CANADA Gender Identity Service, CYFS, Toronto | 1 | Same sample as Wood (2013) above. No new data relevant to our research questions. |
| Zucker, K. J., Bradley, S. J., Owen-Anderson, A., Kibblewhite, S. J., Wood, H., Singh, D., & Choi, K. (2012). Demographics, behavior problems, and psychosexual characteristics of adolescents with gender identity disorder or transvestic fetishism. Journal of Sex & Marital Therapy, 38(2), 151–189. | 2012 | CANADA Gender Identity Service, CYFS, Toronto | 1 | Percentage of youth with CBCL and YSR total scores in clinical range was similar to non-GID referred comparison group, higher than non-referred comparison group. |
| Zucker, K. J., Bradley, S. J., Owen-Anderson, A., Singh, D., Blanchard, R., & Bain, J. (2010). Puberty-blocking hormonal therapy for adolescents with gender identity disorder: A descriptive clinical study. Journal of Gay & Lesbian Mental Health, 15(1), 58–82. doi:10.1080/19359705.2011.530574 | 2010 | CANADA Gender Identity Service, CYFS, Toronto | 1 | More likely to recommend puberty blockers for NF than NM, and less likely to recommend for young people with a lower YSR score. |

Key:

* = derived from author's affiliation and description in paper

ADHD: Attention Deficit / Hyperactivity Disorder; ASD: Autism Spectrum Disorder; ASQ: Asperger Syndrome Quotient; CBCL: Child Behavior Check List; CD: Conduct Disorder; DXA: Dual-energy X-ray Absorptiometry; GD: Gender Dysphoria; GID: Gender Identity Disorder; GnRHa: Gonadotropin-releasing hormone agonist; MH: Mental Health; NM / NF: Natal Male / Natal Female; SRS: Social Responsiveness Scale; yrs: years.

Exclusion codes: 1: included ≥18 year olds–no distinct adolescent data; 2: non clinically-verified GD; 3: case study / series; 4: only included <12 year olds; 5: no original data; 6: non-GD population (e.g., LGBTQ); 7: conference proceedings; 0: no useful data–retained for one of the other 2 papers in this series

greater weight to collective findings than is warranted. The samples included here may also represent those most severely affected not only by GD but by poor mental wellbeing more generally. The second paper in this series will focus specifically on the evidence regarding psychological distress / psychiatric comorbidity in young people with GD.

## Conclusion

GD is an area of growing prominence and therefore generates a growing literature. The observed increase in referrals, particularly in NF adolescents, warrants further investigation. Whilst improvements in availability of services and diagnostic practices are likely to have contributed to this, there has undoubtedly been a shift in cultural attitudes leading to gender non-conformity being more acceptable. The role of de-stigmatisation in the experiences of young people with GD and their decisions to seek support should be explored within the context of mental wellbeing more broadly. It is clear that this is a particularly vulnerable population often presenting as psychologically complex cases: without good epidemiological data we cannot begin to elucidate the lived reality of GD and ensure that intervention / support is equitable, appropriate and timely, and minimises harm. This review has been limited by heterogeneity in recording and reporting practices, and by limited representation beyond national, publicly-funded clinical services. Clinical research centres should gather data prospectively on all referrals with full informed consent and document their assessment protocols, treatments and

outcomes. Whole population studies using administrative datasets reporting on GD / gender non-conformity may be necessary to gain a clear understanding of the epidemiology of clinical GD, along with inter-disciplinary research evaluating the lived experience of adolescents with GD.

## Supporting information

**S1 Checklist. PRISMA checklist.** *From*: Page MJ, McKenzie JE, Bossuyt PM, Boutron I, Hoffmann TC, Mulrow CD, et al. The PRISMA 2020 statement: an updated guideline for reporting systematic reviews. BMJ 2021;372:n71. doi: 10.1136/bmj.n71.
(DOCX)

## Acknowledgments

Special thanks to Ingrid Vinsa, Research Nurse and Administrator at the Gillberg Neuropsychiatry Centre, for her invaluable assistance in obtaining full text papers and assistance to CG in supervision of this piece of work.

## Author Contributions

**Conceptualization:** Lucy Thompson, Christopher Gillberg.

**Data curation:** Lucy Thompson, Darko Sarovic, Philip Wilson, Angela Sämfjord.

**Formal analysis:** Darko Sarovic.

**Methodology:** Lucy Thompson, Darko Sarovic, Philip Wilson, Angela Sämfjord, Christopher Gillberg.

**Project administration:** Lucy Thompson.

**Supervision:** Christopher Gillberg.

**Validation:** Darko Sarovic, Philip Wilson, Angela Sämfjord, Christopher Gillberg.

**Visualization:** Lucy Thompson.

**Writing – original draft:** Lucy Thompson, Christopher Gillberg.

**Writing – review & editing:** Lucy Thompson, Darko Sarovic, Philip Wilson, Angela Sämfjord, Christopher Gillberg.

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
