## [Decision Letter · Decision Letter 0]

29 Sep 2021

PGPH-D-21-00213

A PRISMA systematic review of adolescent gender dysphoria literature: 1) epidemiology

Dear Dr. Thompson,

Thank you for submitting your manuscript to PLOS Global Public Health. After careful consideration, we feel that it has merit but does not fully meet PLOS Global Public Health’s publication criteria as it currently stands. Therefore, we invite you to submit a revised version of the manuscript that addresses the points raised during the review process.

We look forward to receiving your revised manuscript.

Kind regards,

Runsen Chen

Academic Editor

Journal Requirements:

1. We ask that a manuscript source file is provided at Revision. Please upload your manuscript file as a .doc, .docx, .rtf or .tex. If you are providing a .tex file, please upload it under the item type ‘LaTeX Source File’ and leave your .pdf version as the item type ‘Manuscript’.

2. Please provide  separate figure files in .tif or .eps format only and remove any figures embedded in your manuscript file.  Please ensure that all files are under our size limit of 20MB.  

For more information about how to convert your figure files please see our guidelines: Once you've converted your files to .tif or .eps, please also make sure that your figures meet our format requirements

.

Additional Editor Comments (if provided):

Please address the comments from reviewers. 

Reviewers' comments:

Reviewer's Responses to Questions

**Comments to the Author**

1. Does this manuscript meet PLOS Global Public Health’s publication criteria? Is the manuscript technically sound, and do the data support the conclusions? The manuscript must describe methodologically and ethically rigorous research with conclusions that are appropriately drawn based on the data presented.

Reviewer #1: Partly

Reviewer #2: Yes

Reviewer #3: Yes

2. Has the statistical analysis been performed appropriately and rigorously?

Reviewer #1: N/A

Reviewer #2: Yes

Reviewer #3: Yes

3. Have the authors made all data underlying the findings in their manuscript fully available (please refer to the Data Availability Statement at the start of the manuscript PDF file)?

Reviewer #1: No

Reviewer #2: Yes

Reviewer #3: No

4. Is the manuscript presented in an intelligible fashion and written in standard English?

Reviewer #1: Yes

Reviewer #2: Yes

Reviewer #3: Yes

5. Review Comments to the Author

Reviewer #1: In this current manuscript, Thompson et al. presented a A PRISMA systematic review of adolescent gender dysphoria literature as the first part of their three-part systematic review. The authors found no enough evidence to yield the prevalence of gender dysphoria given the current data compiled. In the contrast, they found an increase in the frequency of service utilization, a prominence of assigned female at birth yet no data on the age of onset. The manuscript is well-structured, methods and results are correct. However, I have some concerns for authors' consideration.

Major

1. The searched timeframe in different databases are varied (“we searched Ovid Medline 1946 – October week 4 2020, Embase 1947–present (updated daily), CINAHL 1983–2020, and PsycInfo 1914–2020.), I wonder what could be the reasons the authors chose different start and end time for them. Did they affect the final article selection?

2. The quality of the studies ranged from 45%-96% by CCAT appraisal tool. Did the author consider excluding some poor-quality studies based on the evaluation?

3. Since some of the goals cannot be met given the current evidences/literatures, I recommend the authors delete some unmet goals such as prevalence of GD and age of onset, and adjust the content accordingly. It is not adequate to just discuss it without having enough evidences handy. A follow-up question is whether it’s ready to review GD so thoroughly or should the manuscript be simplified to a narrowed scope?

4. Page 6, methods, study selection part. “Following initial full text screening, all remaining papers were assessed by a second reviewer to reduce the risk of inclusion bias.”. How could the authors be certain for the full inclusion of existing literatures when there was no independent search from at least two authors?

5. Page 6, methods, study selection part. It was mentioned the overlaps were assessed, did they fully or partially overlapped? Please provide more details of the evaluation and extent of overlapping if possible.

6. One page 9, paragraph 2. How could the age of onset range from 1 to 15 year old? I doubt if someone can be diagnosed with GD at the age of 1.

7. Page 10, paragraph 3. How did the author deal with the unclear definition of “referral”? what about the referral could vary in different countries?

8. In terms of the searching strategies. In Table 1. Could the authors explain the different search terms applied to different databases? Why didn’t the authors do a hand search? How did the authors pick databases to search? There are also some major archives such as PubMed and Scopus that weren’t included.

minor

1. figure 1. please fill in the number for right upper frame (additional records…). So as fixing the records excluded #1 and specifying each items under full-text articles excluded.

2. In figure 2, questions 3a has zero literature, is it worth keeping it?

Reviewer #2: Thank you for the opportunity to review this paper. Prevalence of gender dysphoria among adolescents represents an important area of research. I congratulate the authors on their effort and must agree with their overall conclusions that the data on prevalence (or incidence) of GD are simply not available. I have no concerns with the methods of systematic review however I do have one substantive suggestion on how to make this paper more impactful. Given the scant amount of good actionable data I would prefer to see this paper as a call to action and perhaps even a series of specific proposals on how one should go about addressing the question of GD prevalence in the adolescent populations.

Reviewer #3: In PGPH-D-21-00213, the authors seek apply a rigorous framework to produce a systematic review of the epidemiology of gender dysphoria (GD). The topic is timely, and the review appears to be thorough and carefully conducted, applying established criteria for systematic reviews, as well as quality assessment of the included studies. Although the authors do not offer a definitive prevalence estimate for GD, the reasons for this are simple—the requisite studies for a high-quality estimate of this sort have not been done. There is however, a documented shift in the sex ratio of individuals who are seeking treatment for GD, with an increasing number of natal females seeking these services in recent years. My overall impression of the manuscript is positive. The review has been carefully and competently done, and the authors have appropriately framed and contextualized their findings. I especially appreciate that the authors have focussed on instances where GD is verifiable (i.e., assessed by a clinician in a formal framework), rather than relying only on self-reported identification as being transgender. The high bar that the authors set for inclusion in their systematic review is a strength of the manuscript, although it substantially reduces the number of eligible papers. I have included some comments below which I hope are helpful to the authors.

1) Abstract. The authors employ the term “natal gender.” It might be more parsimonious to consistently refer to natal sex (male/female), which is both immutable and for all intents and purposes, binary (despite low frequency of DSD in the population). The use of the term “natal gender” occurs again on page 6, for example, when describing what information was recorded from reports included in the review, as well as in Table 2, and on page 9 (Discussion). Similarly, the term “natal gender ratio” is used on page 8 (although I suspect this terminology is from the paper cited, being specific about a natal sex ratio might be advisable).

2) Introduction. A bit more terminological consistency would great aid readability of the entire manuscript. For example, the authors state, “There has been a parallel increase in demand for interventions to transition from one gender to another, particularly from female to male.” Here, sex and gender terms seem to be somewhat mixed. An individual cannot change their sex (chromosomes, genitalia, or likely most relevant, their gametes), although we can make social decisions to recognize that not all male individuals are men, and not all female individuals are women. As the authors note in the next paragraph, sex and gender are not synonymous. This differentiation may be a good place to introduce terminological consistency, wherein male/female always refer to sex, whereas boy/girl and man/woman are only used to refer to gender. I understand that this may deviate from the typical use in many contexts (legal, medical, etc.), but I think the more that such differentiated terminology is employed, the more it (hopefully) diffuses into dialogues about these subjects. Last, as a sidenote, I especially appreciate the author using the term “sex documented at birth” for the simple reason that medical professionals are rarely wrong about observing/documenting the biological sex of a newborn. Similarly, I am glad to see the authors use the terms “natal female” and “natal male.” These terms are accurate, precise, and indeed less cumbersome than many proposed alternatives.

3) Introduction. The authors use the term “imbalance” between sex and gender for some individuals. Discordance may be a more appropriate terminological choice.

4) The authors note a lack of systematic reviews regarding the epidemiology of GD. That said, I think it is important to acknowledge that some attempts at such reviews (e.g., Zucker, 2017; https://www.publish.csiro.au/sh/sh17067) have been undertaken, and do give some indication of both prevalence, and the trends that the authors are commenting on here.

5) Characterizing the current discussion surrounding GD (and related issues) as an “intense international debate” is perhaps one of the greatest understatements I’ve ever seen in a manuscript. I want to commend the authors on their brave efforts to confront this topic head-on, and bring important data to bear on these discussions.

6) Regarding data from the Netherlands, so far as I know the Amsterdam clinic is the only gender identity service in the Netherlands. This means that the majority of adolescent GD cases in the Netherlands are highly likely to come into contact with the Amsterdam clinic. Obviously, some adolescents with GD may seek alternative services, but at the very least a lower boundary for the incidence/prevalence of GD can be estimated based on the number of adolescents seen by the Amsterdam clinic, as a proportion of all adolescents of that birth year (Statistics Netherlands—https://www.cbs.nl/en-gb—keeps fabulous, population level statistics on broad demographics within the population going back to roughly the 1950s). A similar argument might apply to the London clinic, although the UK has a considerably larger population than the Netherlands, making such estimates unwieldy and inaccurate in the absence of population level data, or valid random sampling. When discussing the prevalence of adolescent GD, the authors simply state that a precise estimate cannot be offered because population samples are not available. Surely some estimate can be given, especially in light of the authors speculating that there may be an increase in adolescents presenting for treatment.

7) The authors mention a supplementary table detailing characteristics of excluded studies, but this supplement does not appear to be included in the reviewer-accessible submission files. This is important, as the authors state directly (page 10) that population-based survey data were excluded from the present sample.

8) I want to commend the authors on their attention to detail, and scientific caution when making conclusions based on the available evidence. Not all scholars studying GD are so careful.

6. PLOS authors have the option to publish the peer review history of their article (what does this mean?). If published, this will include your full peer review and any attached files.

---

## [Decision Letter · Decision Letter 1]

1 Dec 2021

PGPH-D-21-00213R1

A PRISMA systematic review of adolescent gender dysphoria literature: 1) epidemiology

Dear Dr. Thompson,

Thank you for submitting your manuscript to PLOS Global Public Health. After careful consideration, we feel that it has merit but does not fully meet PLOS Global Public Health’s publication criteria as it currently stands. Therefore, we invite you to submit a revised version of the manuscript that addresses the points raised during the review process.

We look forward to receiving your revised manuscript.

Kind regards,

Runsen Chen

Academic Editor

Journal Requirements:

Additional Editor Comments (if provided):

Reviewers' comments:

Reviewer's Responses to Questions

**Comments to the Author**

1. If the authors have adequately addressed your comments raised in a previous round of review and you feel that this manuscript is now acceptable for publication, you may indicate that here to bypass the “Comments to the Author” section, enter your conflict of interest statement in the “Confidential to Editor” section, and submit your "Accept" recommendation.

Reviewer #1: All comments have been addressed

Reviewer #2: All comments have been addressed

Reviewer #3: All comments have been addressed

2. Does this manuscript meet PLOS Global Public Health’s publication criteria? Is the manuscript technically sound, and do the data support the conclusions? The manuscript must describe methodologically and ethically rigorous research with conclusions that are appropriately drawn based on the data presented.

Reviewer #1: Partly

Reviewer #2: Yes

Reviewer #3: Yes

3. Has the statistical analysis been performed appropriately and rigorously?

Reviewer #1: Yes

Reviewer #2: Yes

Reviewer #3: Yes

4. Have the authors made all data underlying the findings in their manuscript fully available (please refer to the Data Availability Statement at the start of the manuscript PDF file)?

Reviewer #1: Yes

Reviewer #2: Yes

Reviewer #3: Yes

5. Is the manuscript presented in an intelligible fashion and written in standard English?

Reviewer #1: Yes

Reviewer #2: Yes

Reviewer #3: Yes

6. Review Comments to the Author

Reviewer #1: I appreciate the authors' great efforts in reporting such valuable topics. There is scarcity of review in GD. But I have a few concerns regarding the revised manuscript.

1. handsearch was not conducted in this review, though it may have little effect on the final results, it still affect the stringency.

2. this review included some obvious problematic data (eg. GD at the age of 1), I would recommend further confirmation with authors of original studies or remove them, which will theoretically impact the results.

3. given the mixed quality of current studies included, the necessity of a systematic review should be revisited. It would be great to see if there are any new updates.

Reviewer #2: I am pleased with the revised version. Thank you.

Reviewer #3: Please see uploaded document.

7. PLOS authors have the option to publish the peer review history of their article (what does this mean?). If published, this will include your full peer review and any attached files.

**Do you want your identity to be public for this peer review?** For information about this choice, including consent withdrawal, please see our Privacy Policy.

Reviewer #1: No

Reviewer #2: No

Reviewer #3: No

---

## [Decision Letter · Decision Letter 2]

3 Feb 2022

A PRISMA systematic review of adolescent gender dysphoria literature: 1) epidemiology

PGPH-D-21-00213R2

Dear Dr Thompson,

We are pleased to inform you that your manuscript 'A PRISMA systematic review of adolescent gender dysphoria literature: 1) epidemiology' has been provisionally accepted for publication in PLOS Global Public Health.

Best regards,

Runsen Chen

Academic Editor

Reviewer Comments (if any, and for reference):

Reviewer's Responses to Questions

**Comments to the Author**

1. If the authors have adequately addressed your comments raised in a previous round of review and you feel that this manuscript is now acceptable for publication, you may indicate that here to bypass the “Comments to the Author” section, enter your conflict of interest statement in the “Confidential to Editor” section, and submit your "Accept" recommendation.

Reviewer #1: All comments have been addressed

Reviewer #3: All comments have been addressed

2. Does this manuscript meet PLOS Global Public Health’s publication criteria? Is the manuscript technically sound, and do the data support the conclusions? The manuscript must describe methodologically and ethically rigorous research with conclusions that are appropriately drawn based on the data presented.

Reviewer #1: Yes

Reviewer #3: Yes

3. Has the statistical analysis been performed appropriately and rigorously?

Reviewer #1: Yes

Reviewer #3: Yes

4. Have the authors made all data underlying the findings in their manuscript fully available (please refer to the Data Availability Statement at the start of the manuscript PDF file)?

Reviewer #1: No

Reviewer #3: No

5. Is the manuscript presented in an intelligible fashion and written in standard English?

Reviewer #1: Yes

Reviewer #3: Yes

6. Review Comments to the Author

Reviewer #1: All questions were addressed. Pleasure to review this manuscript.

Reviewer #3: The authors have taken very seriously the comments of myself and the other reviewer. The changes strengthen the manuscript, and I can now endorse the publication without delay. I want to thank the authors once again for their ambitious and important work, and for responding to my questions, concerns, and critiques with collegiality and poise. Although the answer to the broad question the work aimed to address is perhaps unsatisfying—we simply don't have good prevalence estimates of adolescent gender dysphoria—this is certainly not because the work is lacking in quality. Instead, I hope this impressive review serves as a useful call for more precise and rigorous data on this important question. Many in this area are flying blind, and it serves neither individuals with GD, clinicians trying to establish best practices, or scientists trying to understand this area. Congratulations, and all the best.

7. PLOS authors have the option to publish the peer review history of their article (what does this mean?). If published, this will include your full peer review and any attached files.

**Do you want your identity to be public for this peer review?** For information about this choice, including consent withdrawal, please see our Privacy Policy.

Reviewer #1: No

Reviewer #3: No
